# SO(2) and O(2) Equivariance in Image Recognition with Bessel-Convolutional Neural Networks

## Abstract

For many years, it has been shown how much exploiting equivariances can be beneficial when solving image analysis tasks. For example, the superiority of convolutional neural networks (CNNs) compared to dense networks mainly comes from an elegant exploitation of the translation equivariance. Patterns can appear at arbitrary positions and convolutions take this into account to achieve translation invariant operations through weight sharing. Nevertheless, images often involve other symmetries that can also be exploited. It is the case of rotations and reflections that have drawn particular attention and led to the development of multiple equivariant CNN architectures. Among all these methods, Bessel-convolutional neural networks (B-CNNs) exploit a particular decomposition based on Bessel functions to modify the key operation between images and filters and make it by design equivariant to all the continuous set of planar rotations. In this work, the mathematical developments of B-CNNs are presented along with several improvements, including the incorporation of reflection and multi-scale equivariances. Extensive study is carried out to assess the performances of B-CNNs compared to other methods. Finally, we emphasize the theoretical advantages of B-CNNs by giving more insights and in-depth mathematical details.

## 1 Introduction

For years now, convolutional neural networks (CNNs) are known to be the most powerful tool that we have for image analysis. Their efficiency compared to classic multi-layers perceptrons (MLPs) mainly comes from an elegant exploitation of the translation equivariance involved in image analysis tasks. Indeed, CNNs exploit the fact that patterns can arise at different positions in images by sharing the weights over translations thanks to convolutions. The translation equivariance can be seen as a particular form of prior knowledge and weights can be saved compared to an MLP architecture with similar performances.

By building on the success of exploiting translation equivariance in image analysis, we advocate here that generalizing this to other types of appropriate symmetries can also be useful. For example, in biomedical or satellite imaging, objects of interest can appear at arbitrary positions with arbitrary orientations. To illustrate this, Figure 1 shows four versions of the exact same galaxy that are equally plausible images that could occur in the data set. If the task is to determine the morphology of the galaxy, it is relevant to want these images to be processed in the exact same way. Therefore, introducing rotation equivariance will lead to a more optimal use of the weights and to a better overall efficiency of the models. Being able to guarantee rotation equivariance is also useful to put more trust into models. For instance, experts would be more confident in models that extract the exact same latent features for an object, no matter its particular orientation (of course, depending on the application).

Still, introducing other types of equivariance in an efficient way in CNNs is not straightforward. On the one hand, many works propose brute-force solutions like (i) considerably increasing the training set (data augmentation), or (ii) artificially multiplying the number of filters by directly applying the desired symmetries onto them. In practice, these solutions will lead both to an increase of the training time and the size of the models. Furthermore, most of these methods do not provide any mathematical guarantee regarding the equivariance. On the other hand, a few works propose solutions to efficiently bring more general equivariances

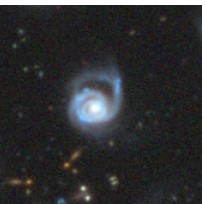 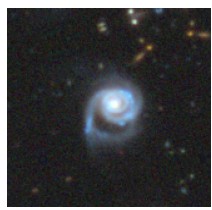 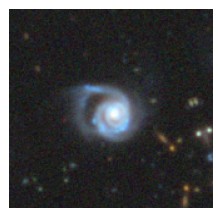 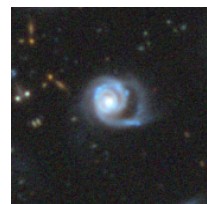

Figure 1: Four rotated versions of the same galaxy image, retrieved from the Galaxy Zoo data set by Willett et al. (2013). This data set contains images of galaxies as well as their morphologies, according to experts. For this application, the orientation is arbitrary and does not contain any information.

in CNNs while providing mathematical guarantees. Bessel-convolutional neural networks (B-CNNs) are one of those, and rely on a particular representation of images that is more convenient to deal with rotations and reflections.

In this work, improvements of B-CNNs compared to the prior work of Delchevalerie et al. (2021) are presented, which are mainly an extension to $O(2)$ equivariance and an optimal choice for the initial $\nu_{\max}$ and $j_{\max}$ meta-parameters based on the Nyquist sampling theorem. Also, we present how multi-scale equivariance can easily be achieved in B-CNNs. Finally, a more extensive study is performed to assess the performances of B-CNNs compared to other state-of-the-art methods on different data sets. To do so, we present the full mathematical developments of B-CNNs and we give more detailed explanations. The advantage of using B-CNNs regarding both the size of the model and the training set is highlighted, along with theoretical and experimental evidence of the equivariance. Our implementation is available online at (*removed for double-blind review, an anonymized version is available as supplementary materials for the reviewers*) along with the scripts used to reproduce the experiments.

## 2  Background and Definition of Invariance and Equivariance

Invariance and equivariance are two different notions that need to be clearly defined for the next sections. Let $\Psi(x, y)$ be an image where $x$ and $y$ represent the pixel coordinates, $K(x, y)$ be an arbitrary filter and $\mathcal{G}$ be a set of transformations that can be applied on the image. The operation defined by $*$ is $\mathcal{G}$-invariant if $(g\Psi)(x, y) * K(x, y) = \Psi(x, y) * K(x, y), \forall g \in \mathcal{G}$, and $\mathcal{G}$-equivariant if $(g\Psi)(x, y) * K(x, y) = g(\Psi * K)(x, y), \forall g \in \mathcal{G}$. In other words, invariance means that the results will be exactly the same for all transformations $g$ of the input image, while equivariance means that the results will be also transformed by the action of $g$.

Convolutional Neural Networks (CNNs) work by applying a succession of convolutions between an input image $\Psi(x, y)$ and some filters. If $K(x, y)$ is one of those particular filters, convolutions are expressed by

$$\Psi(x, y) * K(x, y) = \int_{-R}^{R} \int_{-R}^{R} \Psi(x - x', y - y') K(x', y') \, dx' dy',$$

where $R$ defines the size of the filter. One can now show that CNNs exhibit a translation equivariance (that is, patterns are detected the same way regardless of their particular positions). Indeed, if $\mathcal{T}_{u,v}$ is a translation operator such that it translates the image by an amount of pixels $(u, v)$

$$\mathcal{T}_{u,v} \Psi(x, y) = \Psi(x + u, y + v),$$

one can show that

$$
\begin{aligned}
\mathcal{T}_{u,v}\left(\Psi * K\right)(x,y) &= \int_{-R}^{R}\int_{-R}^{R}\Psi\left((x-x')+u,(y-y')+v\right)K\left(x',y'\right)dx'dy' \\
&= \int_{-R}^{R}\int_{-R}^{R}\Psi\left((x+u)-x',(y+v)-y'\right)K\left(x',y'\right)dx'dy' \\
&= \int_{-R}^{R}\int_{-R}^{R}\mathcal{T}_{u,v}\Psi\left(x-x',y-y'\right)K\left(x',y'\right)dx'dy' \\
&= \left(\mathcal{T}_{u,v}\Psi\right)(x,y) * K\left(x,y\right),
\end{aligned}
$$

which matches the definition of $\mathcal{G}$-equivariance defined earlier, and therefore proves the translation equivariance of CNNs. Figure 2 illustrates this equivariance.

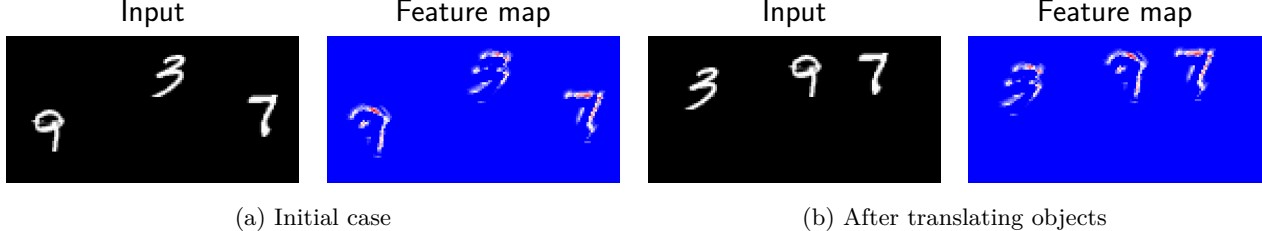

(a) Initial case                                            (b) After translating objects

Figure 2: Illustration of the translation equivariance in CNNs. Both (a) and (b) are made of the same objects, but at different positions. Nevertheless, CNNs process objects the same way independently of their particular absolute positions.

However, regarding other types of transformations, the equivariance in CNNs is generally not achieved. One can for example consider rotations, by defining $R_\alpha$ as an operator that applies a rotation of an angle $\alpha$, such that

$$
R_\alpha\Psi\left(x,y\right) = \Psi\left(x\cos\alpha - y\sin\alpha, x\sin\alpha + y\cos\alpha\right).
$$

By applying a similar development, it clearly appears that

$$
R_\alpha\left(\Psi * K\right)(x,y) \neq \left(R_\alpha\Psi\right)(x,y) * K\left(x,y\right).
$$

This is expected as convolution can be seen as element-wise multiplications with a sliding window, and the result of element-wise multiplications depend on the particular orientation of the matrices. This lack of rotation equivariance will be illustrated in Figure 8a.

## 3 Related Works

Many techniques propose to bring more general equivariance in convolutional neural networks (CNNs). A particular interest was taken in satisfying $SO(2)$ and $O(2)$ equivariance as it is an interesting prior for many applications in image recognition; see for example Chidester et al. (2019) for medical imaging, Dieleman et al. (2015) for astronomical imaging, Li et al. (2020) for satellite imaging and Marcos et al. (2016) for texture recognition. $SO(2)$ is called the special orthogonal group and contains the continuous set of planar rotations, while $O(2)$ is called the orthogonal group and also add all the planar reflections. The different proposed methods can be categorized in different groups: (i) methods that only increase robustness to planar transformations without mathematical guarantees of equivariance, (ii) methods that bring some mathematical guarantees but only for a discrete set of planar transformations (as for example, cyclic $C_n$ and dihedral $D_n$ groups), and (iii) methods that bring mathematical guarantees for the continuous set of transformations.

The most famous technique from the first category is data augmentation (Quiroga et al., 2018). While robustness can be considerably increased with data augmentation, it still requires for the model to learn the

equivariance, as it is not used as an explicit constraint. No theoretical guarantees can then be provided, and extracted features will generally not be the same for rotated versions of a particular object. Next to data augmentation, one can also cite spatial transformer networks by Jaderberg et al. (2015), rotation invariant and Fisher discriminative CNNs by Cheng et al. (2016), deformable CNNs by Dai et al. (2017), and SIFT-CNNs by Kumar et al. (2018). The main drawback of such methods lies in the fact that, as models still learn the equivariance by themselves, many parameters are used to encode redundant information. Therefore, it leads to methods of category (ii) that aim to make models equivariant to discrete groups like $C_n$ or $D_n$. One can for example cite Group-CNNs by Cohen & Welling (2016), deep symmetry networks by Gens & Domingos (2014), steerable CNNs by Cohen & Welling (2017), steerable filter CNNs by Weiler et al. (2018), dense steerable filter CNNs by Graham et al. (2020), spherical CNNs by Cohen et al. (2018) and Deformation Robust Roto-Scale-Translation Equivariant CNNs by Gao et al. (2021). Compared to category (i), equivariance to a finite number of planar transformations is generally obtained by tying the weights for several transformed versions of the filter. Nevertheless, even if guarantees are now obtained, it is only for a finite set of transformations and it still involves computations with many parameters to encode the equivariance (for example, $5 \times 5$ filters in a $D_8$-invariant convolutional layer will be made of $5 \times 5 \times 8 \times 2 = 400$ parameters[1]). Finally, for the third category (iii), one can cite general $E(2)$-equivariant steerable CNNs ($E(2)$-CNNs) by Weiler & Cesa (2019), where equivariance to continuous groups can be obtained by using a finite number of irreducible representations, harmonic networks (HNets) by Worrall et al. (2017) that use spherical harmonics to achieve a rotational equivariance by maintaining a disentanglement of rotation orders in the network, and Finzi et al. (2020) who generalize equivariance to arbitrary transformations from Lie groups. However, authors of $E(2)$-CNNs highlight that approximating $SO(2)$ (resp, $O(2)$) by using $C_n$ (resp, $D_n$) groups instead of using a finite number of irreducible representations leads to better results. It follows that $E(2)$-equivariant CNNs are most of the time equivalent to methods of category (ii). Regarding HNets, they are only $SO(2)$ equivariant and involve complex values in the network that are poorly compatible with many already existing tools (for example, activation functions and batch normalization layers should be adapted, saliency maps cannot be easily computed, etc.). Finally, Esteves et al. (2018) introduce Polar Transformer Networks that use a log-polar representation of images. In this representation, rotations around the origin become vertical shifts and dilations become horizontal shifts. Therefore, the translation equivariance of CNNs is converted to a rotation and scaling one. However, this approach only handles global rotation equivariance (not a local one at the scale of the kernel) and requires to have a global origin for the rotations, which makes this technique quite different from the other ones.

Recently, another type of equivariant CNNs also emerged. While symmetries can be seen as a user constraint for all the previously mentioned techniques, these new equivariant CNNs architectures find by themselves during the training phase the symmetries that should be considered. One can for example cite the work of Dehmamy et al. (2021) in this direction. This is particularly useful when users do not know and have no insight about the symmetries that can be involved in data, or when symmetries are unexpected. However, the aim of such methods differs from the previous ones because symmetries are no longer applied as constraints. Therefore, those methods rely more on the training data, and are useful in a different context of applications. A discussion about the strengths and weaknesses of these methods compared to others is provided at the end of the paper, in Section 8.

Our work is a direct follow-up of the prior work of Delchevalerie et al. (2021), which built on the use of Bessel functions in order to propose a new method that belongs to the third category. One should also cite the work of Cheng et al. (2019) who propose to use a Fourier-Bessel decomposition of the filters to both reduce model size and introduce group equivariance. Compared to the state of the art, Bessel-convolutional neural networks (B-CNNs) initially proposed a new original technique to bring $SO(2)$ equivariance, while being easy to use with already existing frameworks. Indeed, B-CNNs involve real-valued feature maps and can be expressed in terms of vanilla convolutions. In this work, we emphasize the theoretical advantages of B-CNNs by giving more mathematical details. Also, further improvements compared to the prior work of B-CNNs are presented, as for example by making them $O(2)$ and multi-scale equivariant, and automatically inferring optimal choices for some meta-parameters. Finally, a more extensive comparative study is also carried out to highlight the strengths and weaknesses of different methods.

---

[1]However, note that only $5 \times 5 = 25$ parameters are learnable as the other ones are just transformed versions of the initial filter.

# 4 Using Bessel Functions in Image Analysis

In Bessel-convolutional neural networks (B-CNNs), Bessel coefficients are used instead of the raw pixel values conventionally used in vanilla convolutional neural networks (CNNs). This section describes the Bessel functions, and how they can be used to compute these Bessel coefficients. Also, some particular properties of Bessel functions and Bessel coefficients are presented. The aim of this section is to give more insights about the reasons that motivate the use of Bessel functions to achieve different kind of equivariance in CNNs. Compared to the work of Delchevalerie et al. (2021), additional mathematical details are provided as well as a discussion on how to perform an optimal choice for the initial meta-parameters $\nu_{\max}$ and $j_{\max}$, and how Bessel coefficients can also be used to express reflections.

## 4.1 Bessel Functions and Bessel Coefficients

Bessel functions are particular solutions of the differential equation

$$x^2 \frac{d^2 y}{dx^2} + x \frac{dy}{dx} + \left(x^2 - \nu^2\right) y = 0,$$

which is known as the Bessel's equation. The solution of this equation can be written as

$$y\left(x\right) = A J_\nu\left(x\right) + B Y_\nu\left(x\right),$$

where $A$ and $B$ are two constants, and $J_\nu\left(x\right)$ and $Y_\nu\left(x\right)$ are called the Bessel functions of the first and second kind, respectively. It has to be noted that these functions are well-defined for orders $\nu \in \mathbb{R}$ in general. In B-CNNs, only the Bessel functions of the first kind are used since $Y_\nu\left(x\right)$ diverges for $x = 0$. Indeed, Bessel functions will be used to express images that can take arbitrary values, including at the origin. Examples of Bessel functions of the first kind for different integer orders $\nu$ can be seen in Figure 3a.

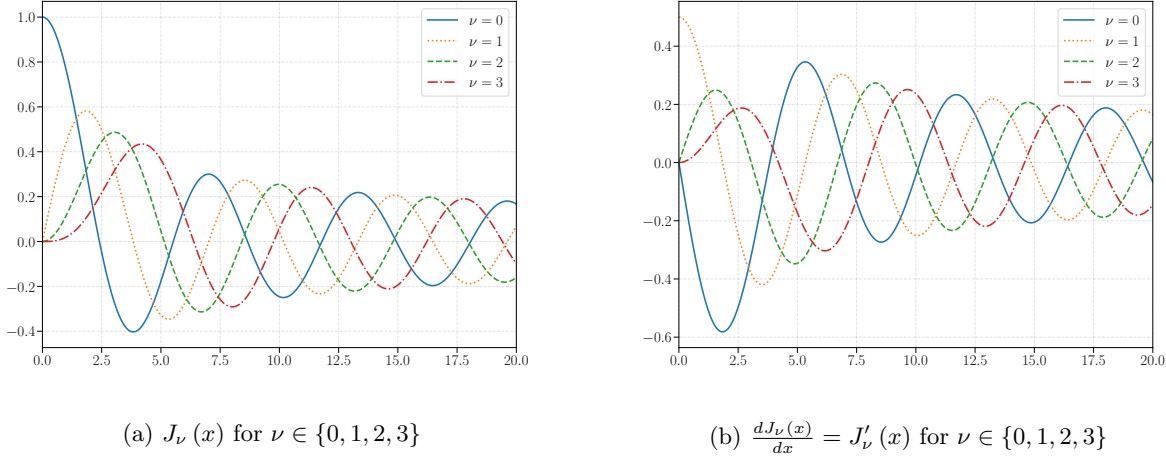

(a) $J_\nu\left(x\right)$ for $\nu \in \{0, 1, 2, 3\}$        (b) $\frac{dJ_\nu\left(x\right)}{dx} = J_\nu'\left(x\right)$ for $\nu \in \{0, 1, 2, 3\}$

Figure 3: Bessel functions of the first kind are presented along with their derivatives for several integer orders $\nu \in \{0, 1, 2, 3\}$.

From a mathematical point of view, Bessel's equation arises when solving Laplace's or Helmholtz's equation in cylindrical or spherical coordinates. Bessel functions are thus particularly well-known in physics as they appear naturally when solving many important problems, mainly when dealing with wave propagation in cylindrical or spherical coordinates (Riley et al., 2006). Since Bessel functions naturally arise when modeling different problems with circular symmetries in physics, these functions are particularly useful to express more conveniently problems with circular symmetries in other domains. This ascertainment motivated the prior work of Delchevalerie et al. (2021) to express images in a particular basis made of Bessel functions of the first kind.

Bessel functions of the first kind can be used to build a particular basis

$$\left\{ N_{\nu,j} J_\nu \left( k_{\nu,j} \rho \right) e^{i\nu\theta}, \forall \nu, j \in \mathbb{N} \right\}, \text{ where } N_{\nu,j} = 1/\sqrt{2\pi \int_0^R \rho J_\nu^2 \left( k_{\nu,j} \rho \right) d\rho}, \tag{4.1}$$

for the representation of images defined in a circular domain of radius $R$, where $\rho$ and $\theta$ are the polar coordinates (the Euclidean distance from the origin and the angle with the horizontal axis, respectively). By carefully choosing $k_{\nu,j}$, this basis can be made orthonormal for all squared-integrable functions $f$ such that $f : D^2 \subset \mathbb{R}^2 \longrightarrow \mathbb{R}$ (where the domain $D^2$ is a disk in $\mathbb{R}^2$). To do so, one can choose $k_{\nu,j}$ such that $J_\nu' \left( k_{\nu,j} R \right) = 0$. The proof for the orthonormality of the basis in this case is presented in Appendix A. Another common choice that also leads to orthonormality is to use $J_\nu \left( k_{\nu,j} R \right) = 0$. Indeed, these two constraints are suitable since a property of the Bessel functions is that $J_\nu' \left( x \right) = \frac{1}{2} \left( J_{\nu-1} \left( x \right) - J_{\nu+1} \left( x \right) \right)$. Therefore, applying the constraint on $J_\nu \left( x \right)$ or on $J_\nu' \left( x \right)$ are both valid solutions that bring orthonormality. However, in our particular case, we choose to apply the constraint on $J_\nu' \left( x \right)$ because it makes it more convenient to represent arbitrary functions, as shown by Mayer & Vigneron (1999). The reason is that it exists a solution $k_{\nu,j} = 0$ for $\nu = 0$ such that $J_\nu' \left( k_{\nu,j} R \right) = 0$, which would not be the case with the constraint based on $J_\nu \left( x \right)$ (see Figure 3). Therefore, the first element in the basis $N_{0,0} J_0 \left( k_{0,0} \rho \right) e^{i0\theta}$ will be equal to $N_{0,0}$. As the result is constant and does not depend on $\rho$ and $\theta$, this element can be used to describe an arbitrary constant intensity in $f$. Figure 4 presents some elements of the basis, including the first one. Also, one can point out that when the order $\nu$ increases, the angular frequency (the number of zeros along the $\theta$-polar-coordinate) of the basis element increases. On the other side, when the order $j$ increases, the radial frequency (the number of zeros along the $\rho$-polar-coordinate) increases.

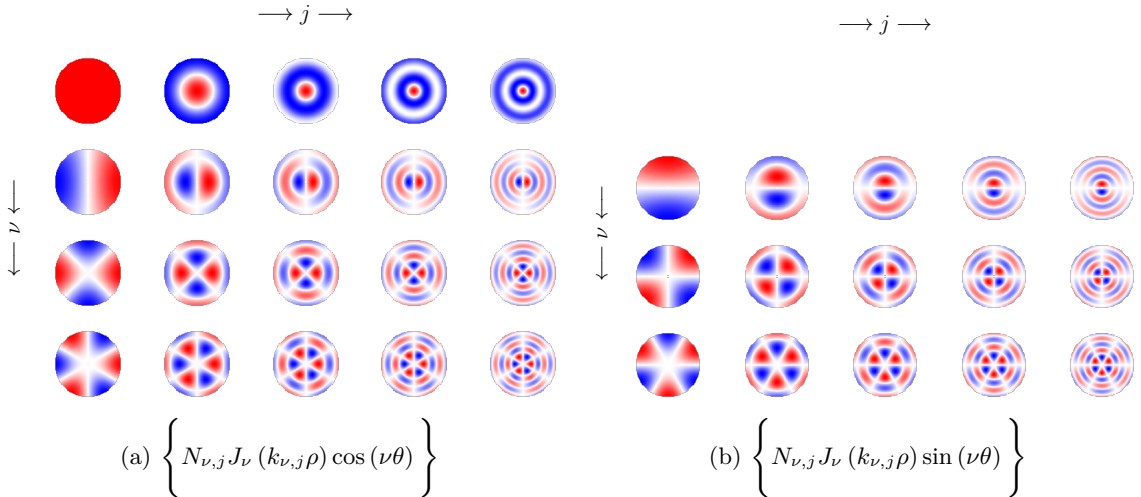

(a) $\left\{ N_{\nu,j} J_\nu \left( k_{\nu,j} \rho \right) \cos \left( \nu\theta \right) \right\}$ (b) $\left\{ N_{\nu,j} J_\nu \left( k_{\nu,j} \rho \right) \sin \left( \nu\theta \right) \right\}$

Figure 4: Real (a) and imaginary (b) parts of the basis described by Equation (4.1) for different $\nu$ and $j$. Red and blue correspond to positive and negative values, respectively. One can see that $\nu$ is linked to an angular frequency, and $j$ to a radial frequency. Note as for $\nu = 0$, there is no imaginary part.

An arbitrary function in polar coordinates $\Psi \left( \rho, \theta \right) : D^2 \subset \mathbb{R}^2 \longrightarrow \mathbb{R}$ can be represented in the basis presented in Equation (4.1) as

$$\Psi \left( \rho, \theta \right) = \sum_{\nu=-\infty}^{\infty} \sum_{j=0}^{\infty} \varphi_{\nu,j} \, N_{\nu,j} J_\nu \left( k_{\nu,j} \rho \right) e^{i\nu\theta}, \tag{4.2}$$

where $\varphi_{\nu,j} \in \mathbb{C}$ are the Bessel coefficients of $\Psi \left( \rho, \theta \right)$. These $\varphi_{\nu,j}$ are the mathematical projection of $\Psi \left( \rho, \theta \right)$ on the Bessel basis. Therefore, they are obtained by

$$\varphi_{\nu,j} = \int_0^{2\pi} \int_0^R \rho \left[ N_{\nu,j} J_\nu \left( k_{\nu,j} \rho \right) e^{i\nu\theta} \right]^* \Psi \left( \rho, \theta \right) d\rho d\theta, \tag{4.3}$$

where the element inside the brackets corresponds to the element $(\nu, j)$ in the Bessel basis. By integrating on $D^2$, it computes the representation of $\Psi(\rho, \theta)$ in this basis.

In B-CNNs, images are represented by a set of those Bessel coefficients instead of directly using the raw pixel values. Further motivations about this will be given later. However, one can already point out that Equation (4.2) needs in principle an infinite number of Bessel coefficients in order to faithfully represent the initial function $\Psi(\rho, \theta)$. From a numerical point of view, these two infinite summations need to be truncated. First of all, one can show that it is not necessary to compute $\varphi_{\nu,j}$ when $\nu$ is a negative integer, since $\varphi_{\nu,j}$ and $\varphi_{-\nu,j}$ are not independent. Indeed, if $\nu \in \mathbb{N}$, $J_\nu(x)$ and $J_{-\nu}(x)$ are linked by the relation

$$J_{-\nu}(x) = (-1)^\nu J_\nu(x).$$ (4.4)

Furthermore, Bessel functions also satisfy

$$J_\nu(-x) = (-1)^\nu J_\nu(x),$$ (4.5)

which means that $J_\nu$ is an even function if $\nu$ is even, and an odd function otherwise. By injecting Equations (4.4) and (4.5) in Equation (4.3), one can show that (proof can be found in Appendix B)

$$\begin{cases} \Re(\varphi_{-\nu,j}) = (-1)^\nu \Re(\varphi_{\nu,j}) \\ \Im(\varphi_{-\nu,j}) = (-1)^{\nu+1} \Im(\varphi_{\nu,j}). \end{cases}$$ (4.6)

The infinite summation for $\nu$ in Equation (4.2) can be decomposed in two summations, one for $\nu \in \{-\infty, \ldots, -1\}$ and another one for $\nu \in \{0, \ldots, \infty\}$. By exploiting the link between $\varphi_{\nu,j}$ and $\varphi_{-\nu,j}$, the infinite summation for $\nu \in \{-\infty, \ldots, \infty\}$ can then be reduced to a summation for $\nu \in \{0, \ldots, \infty\}$, and it is not necessary to compute Bessel coefficients for negative $\nu$ orders. Finally, in order to truncate the infinite summations, two meta-parameters $\nu_{\max}$ and $j_{\max}$ are defined, and the Bessel coefficients are only computed for $\nu$ (resp. $j$) in $\{0, ..., \nu_{\max}$ (resp. $j_{\max})\}$. Nonetheless, it is difficult to make a good choice for these meta-parameters and this may be rather automated by constraining $k_{\nu,j}$ with an upper limit. This is clearly supported by Figure 4, as it shows that high $\nu$ ($j$, respectively) orders correspond to basis elements with an high angular (radial, respectively) frequency. Therefore, as images are sampled on a discrete Cartesian grid, information about frequencies higher than an upper limit cannot be conserved. This upper limit can be determined by the Nyquist frequency, as done by Zhao & Singer (2013), in order to both minimize the aliasing effect and maximize the amount of information preserved by the Bessel coefficients. The aliasing effect occurs when frequencies are numerically mismatched to another frequency information due to an unfortunate sampling. The aliasing effect is illustrated in Figure 5.

To avoid such issues, one should not sample at a frequency higher than the maximal frequency that can theoretically be contained in the image. From now, let us suppose that the radius $R$ of an image is arbitrarily set to 1. If the image is made up of $2n \times 2n$ pixels sampled on a Cartesian grid, it leads to a resolution of $1/n$. Hence, the sampling rate is $n$ and the associated Nyquist frequency (the band-limit) is $n/2$. Therefore, it is optimal to use only the $\varphi_{\nu,j}$ that satisfy the constraint

$$\frac{k_{\nu,j}}{2\pi} \leq \frac{n}{2},$$

because those are the only ones that carry information really contained on the finite Cartesian grid. We then define[2]

$$k_{\max} = \max_{\nu,j} k_{\nu,j} \text{ s.t. } \frac{k_{\nu,j}}{2\pi} \leq \frac{n}{2}.$$ (4.7)

One of the consequences of this constraint is that, for larger $\nu$ orders, a smaller number of Bessel coefficients will be computed, as $k_{\nu,j}$ will reach $k_{\max}$ more rapidly. Indeed, the zeros of $J_\nu'(x)$ (that are the $k_{\nu,j}$'s if $R=1$) are shifted toward higher $x$ values (see the shifting toward the right for $J_\nu'(x)$ when $\nu$ increases in

---

[2]It is interesting to mention that this constraint is also a common choice in numerical physics, where $\frac{k}{2\pi} = \frac{1}{\lambda}$, $\lambda$ being the wavelength. It is meaningless to use larger values for $k$, as it corresponds to wavelengths smaller than the resolution of space.

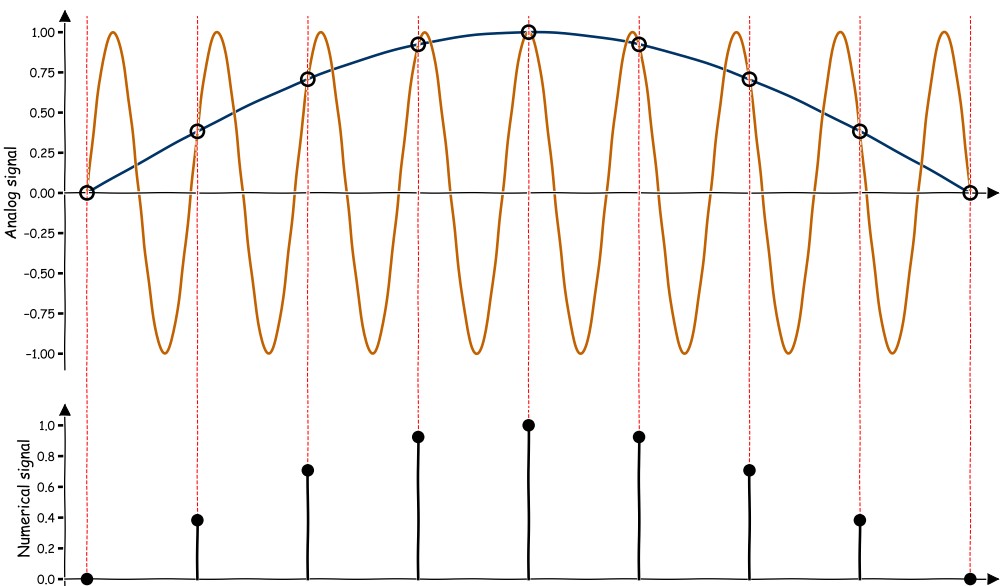

Figure 5: Illustration of the aliasing effect. From a numerical point of view, both analog signals in the upper part can be interpreted in the same way if the sampling rate (indicated by the red dashed lines) is fixed without considering it.

Figure 3b). To conclude this section, the function $\Psi(\rho, \theta)$ will be represented by a matrix with the general form

$$
\begin{pmatrix}
\varphi_{0,0} & \cdots & \varphi_{0,j} & \cdots & \varphi_{0,j_{\max}} \\
\vdots & \ddots & \vdots & \ddots & \vdots \\
\varphi_{\nu,0} & \cdots & \varphi_{\nu,j} & \cdots & 0 \\
\vdots & \ddots & \vdots & \ddots & \vdots \\
\varphi_{\nu_{\max},0} & \cdots & 0 & \cdots & 0
\end{pmatrix},
$$

where each non-zero element corresponds to values for $\nu$ and $j$ that satisfy $k_{\nu,j} \le k_{\max}$. Figure 6 presents an example where $\Psi(\rho, \theta)$ is an arbitrary image. Bessel coefficients are computed in this particular case with Equation (4.3), and the inverse transformation described by Equation (4.2) is also performed for different thresholds ($k_{\nu,j} \le \frac{1}{2}k_{\max}$, $k_{\nu,j} \le k_{\max}$ and $k_{\nu,j} \le \frac{4}{3}k_{\max}$) to check how much information is preserved by the Bessel coefficients and how the thresholds modify the information content. It also illustrates the aliasing effect that appears when considering high-frequency information that may not be initially present in the image.

## 4.2 Effect of Rotations

To understand why using Bessel coefficients is more convenient than using raw pixel values, one can determine the consequence of a rotation of $\Psi(\rho, \theta)$ on $\varphi_{\nu,j}$. Let $\Psi^{\mathrm{rot}}(\rho, \theta)$ be the rotated version of $\Psi(\rho, \theta)$ for an angle $\alpha \in [0, 2\pi[$, that is, $\Psi^{\mathrm{rot}}(\rho, \theta) = \Psi(\rho, \theta - \alpha)$. Its Bessel coefficients are given by

$$
\varphi_{\nu,j}^{\mathrm{rot}} = \int_0^{2\pi} \int_0^R \rho \left[ N_{\nu,j} J_\nu(k_{\nu,j}\rho) e^{i\nu\theta} \right]^* \Psi^{\mathrm{rot}}(\rho, \theta) \, d\rho d\theta.
$$

By defining $\theta' = \theta - \alpha$, it leads to

$$
\varphi_{\nu,j}^{\mathrm{rot}} = \int_0^{2\pi} \int_0^R \rho \left[ N_{\nu,j} J_\nu(k_{\nu,j}\rho) e^{i\nu\theta'} \right]^* \Psi(\rho, \theta') e^{-i\nu\alpha} \, d\rho d\theta' = \varphi_{\nu,j} e^{-i\nu\alpha}. \tag{4.8}
$$

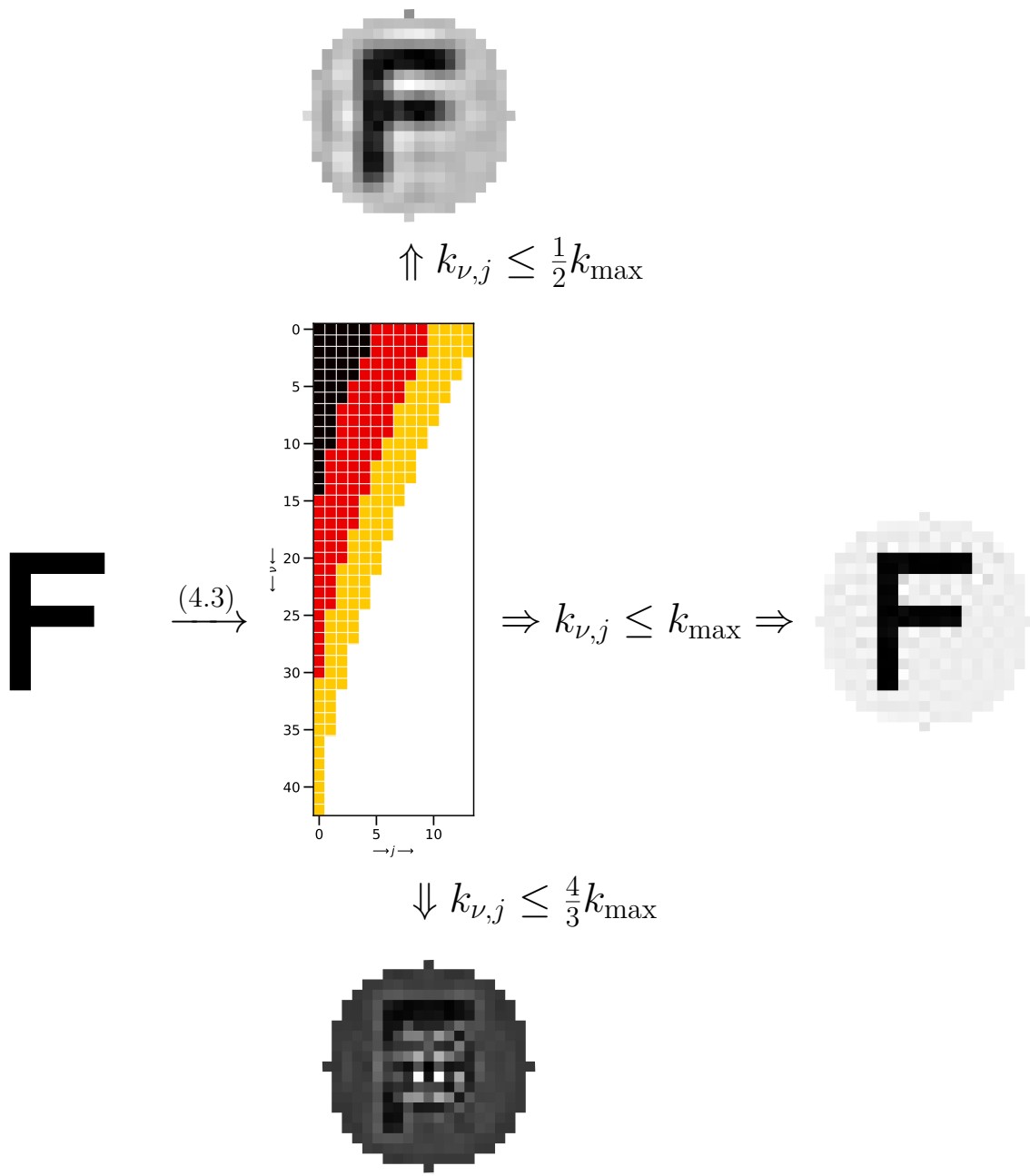

Figure 6: Decomposition and recomposition of an arbitrary image in the Bessel-Fourier domain. The middle part of the image illustrates the Bessel coefficients that are considered for three different reconstruction of the image (each square is a complex-valued Bessel coefficient $\varphi_{\nu,j}$). For the upper part, all the coefficients s.t. $k_{\nu,j} \leq \frac{1}{2}k_{\max}$ (sub-optimal) are considered (■), resulting in a loss of high frequency information. For the lower part, all the coefficients s.t. $k_{\nu,j} \leq \frac{4}{3}k_{\max}$ (over-optimal) are considered (■+■+■), resulting in a reconstruction incorporating misleading high frequencies (aliasing effect). Finally, the optimal choice is presented on the right (■+■).

Therefore, a rotation of an arbitrary function by an angle $\alpha$ only modifies its Bessel coefficients by a multiplication factor $e^{-i\nu\alpha}$. This motivated the development of B-CNNs as it makes rotations conveniently expressed in the Fourier-Bessel transform domain (analogously to how the Fourier transform maps translations to multiplications by complex exponentials). The upper part of Figure 7 illustrates this property (the image is rotated by $\frac{\pi}{2}$ after multiplying its Bessel coefficients by $e^{-i\nu\frac{\pi}{2}}$).

### 4.3 Effect of Reflections

In addition to rotations, Bessel coefficients are also particularly useful when it comes to express reflections. To check this, let $\Psi^{\mathrm{ref}}(\rho,\theta)$ be the reflected version of $\Psi(\rho,\theta)$ along the vertical axis[3]. The Bessel coefficients of $\Psi^{\mathrm{ref}}(\rho,\theta) = \Psi(\rho,\pi-\theta)$ are given by

$$\varphi_{\nu,j}^{\mathrm{ref}} = \int_0^{2\pi}\int_0^R \rho\left[N_{\nu,j}J_\nu\left(k_{\nu,j}\rho\right)e^{i\nu\theta}\right]^*\Psi^{\mathrm{ref}}(\rho,\theta)\,d\rho d\theta.$$

Similarly to what is done for arbitrary rotations, one can define $\theta' = \pi - \theta$. This leads to

$$\varphi_{\nu,j}^{\mathrm{ref}} = -\int_\pi^{-\pi}\int_0^R \rho\left[N_{\nu,j}J_\nu\left(k_{\nu,j}\rho\right)e^{i\nu\left(\pi-\theta'\right)}\right]^*\Psi(\rho,\theta')\,d\rho d\theta'.$$

It is shown in Appendix B that $N_{\nu,j} = N_{-\nu,j}$ and that $k_{-\nu,j} = k_{\nu,j}$. By exploiting this along with Equation (4.4) and the fact that $e^{-i\nu\pi} = \cos(\nu\pi) = (-1)^\nu$, one can show that

$$\begin{aligned}
\varphi_{\nu,j}^{\mathrm{ref}} &= \int_0^{2\pi}\int_0^R \rho\left[N_{-\nu,j}\left(-1\right)^\nu J_{-\nu}\left(k_{-\nu,j}\rho\right)e^{-i\nu\theta'}\right]^* e^{-i\nu\pi}\Psi(\rho,\theta')\,d\rho d\theta' \\
&= \int_0^{2\pi}\int_0^R \rho\left[N_{-\nu,j}J_{-\nu}\left(k_{-\nu,j}\rho\right)e^{-i\nu\theta'}\right]^*\Psi(\rho,\theta')\,d\rho d\theta' \\
&= \varphi_{-\nu,j}.
\end{aligned} \tag{4.9}$$

Therefore, performing a reflection of the image only switches the Bessel coefficients $\varphi_{\nu,j}$ to $\varphi_{-\nu,j}$. Thanks to Equations (4.6), it is equivalent to changing the sign of the real (resp. imaginary) part of the Bessel coefficient if $\nu$ is odd (resp. even). Therefore, in addition to rotations, Bessel coefficients are also really convenient to express reflections. This is illustrated in the lower part of Figure 7, where the image is reflected vertically after switching each $\varphi_{\nu,j}$ with $\varphi_{-\nu,j}$.

## 5 Designing Operations with Bessel Coefficients

In CNNs, the main mathematical operation is a convolutional product between the different filters and the image (or feature maps if deeper in the network). Each filter sweeps the image locally and the weights are multiplied with the raw pixel values. However, in B-CNNs, the aim is to use Bessel coefficients instead of raw pixel values to benefit from the properties described in the previous sections. Yet, the key operation between the parameters of the network (filters) and the images needs to be adapted. This section first presents the mathematical operation used to achieve equivariance under rotation. After that, the initial work of Delchevalerie et al. (2021) is extended to also achieve equivariance under reflection.

### 5.1 A Rotation Equivariant Operation

The convolution performed in CNNs between an arbitrary image $\Psi(x,y)$ and a particular kernel $K(x,y)$ defined for $(x,y) \in [-R,R] \times [-R,R]$ can be written

$$\Psi(x,y) * K(x,y) = \int_{-R}^R\int_{-R}^R \Psi\left(x-x',y-y'\right)K\left(x',y'\right)dx'dy'.$$

---

[3]The reflection along the horizontal axis is not needed, since it can be decomposed as a vertical reflection and a rotation of $\pi$ radians. By composition, reflection equivariance along the horizontal axis is automatically achieved if the layer is equivariant to rotations and reflections along the vertical axis.

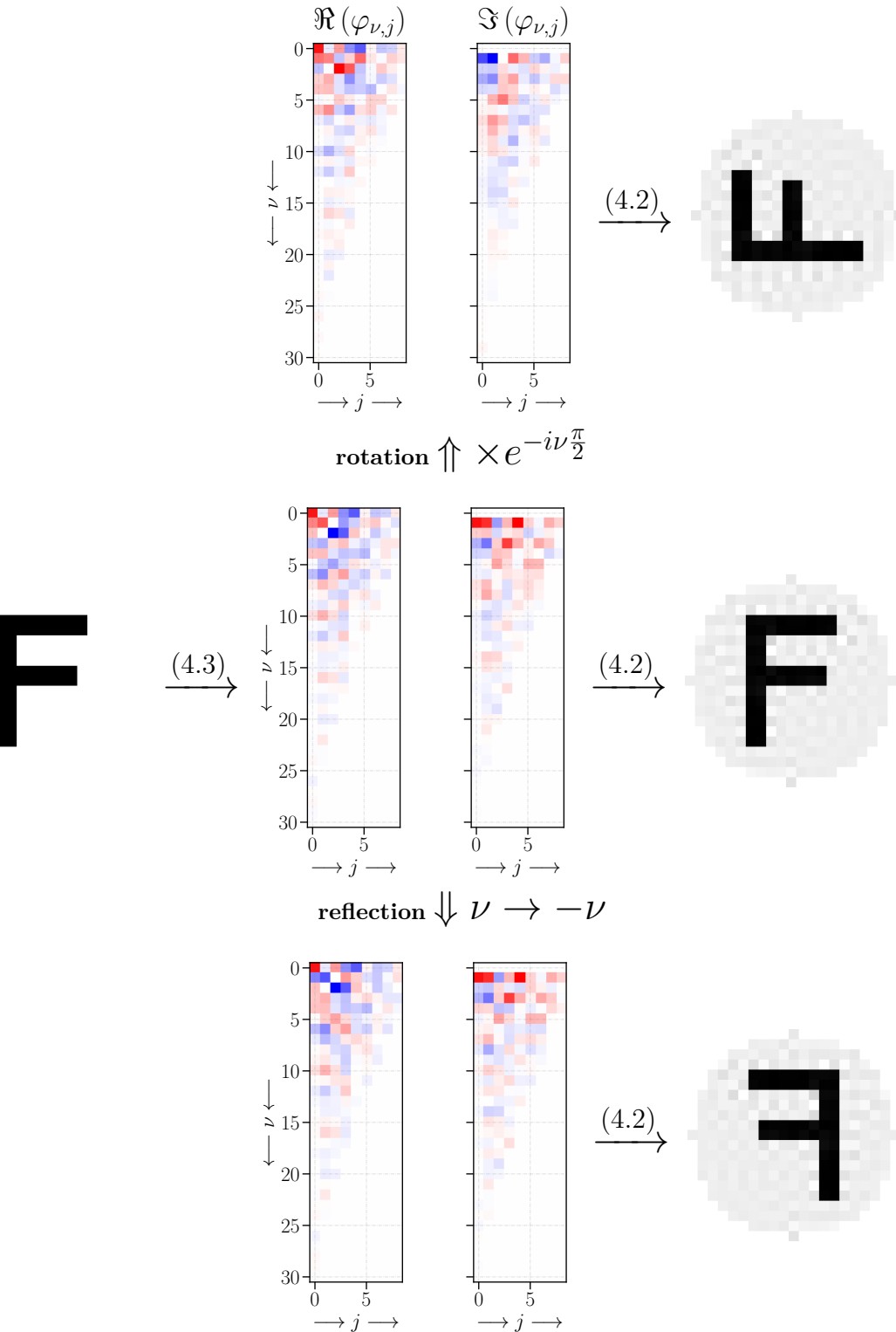

Figure 7: Decomposition of an arbitrary image in some of its Bessel coefficients. Those Bessel coefficients constitute a particular representation of the image and can be used to recover it thanks to Equation (4.2) (middle part). It also illustrates how Bessel coefficients can be conveniently used to apply rotations (upper part) and reflections (lower part).

By defining $\Psi^{(x,y)}(x', y') = \Psi(x - x', y - y')$ and converting the integration from Cartesian to polar coordinates, it leads to

$$\Psi(x, y) * K(x, y) = \int_0^{2\pi} \int_0^R \Psi^{(x,y)}(\rho, \theta) K(\rho, \theta) \rho d\rho d\theta. \tag{5.1}$$

Now, in order to obtain a result that is invariant to the particular orientation of $\Psi^{(x,y)}(\rho, \theta)$, one can decompose it in its Bessel coefficients $\varphi_{\nu,j}^{(x,y)}$ and use Equation (4.8) to implement arbitrary rotations. Next, the idea is to combine this with an integration over $\alpha$ in order to equally consider all the possible orientations of the original image while multiplying it with the kernel, resulting in a rotation invariance. We introduce thus a new rotation invariant convolutional operation $a(x, y) = \Psi(x, y) \star K(x, y)$ described by

$$
\begin{aligned}
a(x, y) &= \frac{1}{2\pi} \int_0^{2\pi} \Big| \int_0^{2\pi} \int_0^R \sum_{\nu,j} \varphi_{\nu,j}^{(x,y)} e^{-i\nu\alpha} N_{\nu,j} J_\nu(k_{\nu,j}\rho) e^{i\nu\theta} K(\rho, \theta) \rho d\rho d\theta \Big|^2 d\alpha \\
&= \frac{1}{2\pi} \int_0^{2\pi} \Big| \sum_{\nu,j} \varphi_{\nu,j}^{(x,y)} e^{-i\nu\alpha} \int_0^{2\pi} \int_0^R \left[ N_{\nu,j} J_\nu(k_{\nu,j}\rho) e^{-i\nu\theta} \right]^* K(\rho, \theta) \rho d\rho d\theta \Big|^2 d\alpha \\
&= \frac{1}{2\pi} \int_0^{2\pi} \Big| \sum_{\nu,j} \varphi_{\nu,j}^{(x,y)} e^{-i\nu\alpha} \kappa_{\nu,j}^* \Big|^2 d\alpha, \tag{5.2}
\end{aligned}
$$

where $\kappa_{\nu,j}$ refers to the Bessel coefficients of the kernel $K(\rho, \theta)$. Thanks to the integration over $\alpha$ from 0 to $2\pi$ and the multiplication of $\varphi_{\nu,j}^{(x,y)}$ by $e^{-i\nu\alpha}$ to describe the effect of rotations, the operation with the kernel is performed for all continuous rotations $\Psi^{(x,y)}(\rho, \theta - \alpha)$ of the original image, where $\alpha \in [0, 2\pi[$. Therefore, $a(x, y)$ should not depend on the particular initial orientation anymore. This idea to achieve a rotation invariance through an integration over an angular dimension has already been explored in non-Euclidean CNNs, like for example in the work of Masci et al. (2015). The operation is still a convolution-like operation as the filters still sweep the input image to progressively construct the feature maps. Therefore, feature maps will be obtained in both a translation and rotation equivariant way. Feature maps in B-CNNs are equivariant because they are obtained by a succession of local invariances (in other words, the key operation between the image and the filter in Equation 5.2 is rotation invariant, but as it is successively performed for local parts of the input, it leads to a global rotation equivariance).

In Equation (5.2), a squared modulus $|\cdot|^2$ is introduced in our operation since without it, one obtains

$$a(x, y) = \frac{1}{2\pi} \sum_{\nu,j} \varphi_{\nu,j}^{(x,y)} \kappa_{\nu,j}^* \int_0^{2\pi} e^{-i\nu\alpha} d\alpha = \sum_j \varphi_{0,j}^{(x,y)} \kappa_{0,j}^*,$$

and only the subset of coefficients $\{\varphi_{0,j}^{(x,y)}, \forall j\}$ will contribute to $a(x, y)$. This subset alone, however, does not constitute a faithful representation of the image. The operation without the squared modulus would therefore inevitably lead to an important loss of information. The factor $\frac{1}{2\pi}$ was introduced finally for normalization purpose.

Computing Equation (5.2) seems not straightforward as it requires to perform a numerical integration. However, one can develop it further in order to obtain an analytical solution, which will be much more convenient to implement in practice. To do so, one can first develop the squared modulus given that

$$
\begin{aligned}
\Big| \sum_{i=1}^k \alpha_i |z_i| e^{i\theta_i} \Big|^2 &= \sum_{m,j} \Re(\alpha_m) \Re(\alpha_j) |z_m z_j| \cos(\theta_m - \theta_j) \\
&+ \sum_{m,j} \Im(\alpha_m) \Im(\alpha_j) |z_m z_j| \cos(\theta_m - \theta_j) \\
&- 2 \sum_{m,j} \Im(\alpha_m) \Re(\alpha_j) |z_m z_j| \sin(\theta_m - \theta_j), \tag{5.3}
\end{aligned}
$$

where $\alpha_i \in \mathbb{C}$ and $z_i = |z_i| e^{i\theta_i} \in \mathbb{C}$. By re-writing the complex valued Bessel coefficients $\varphi_{\nu,j}^{(x,y)}$ as $|\varphi_{\nu,j}^{(x,y)}| e^{i\theta_{\nu,j}}$, it leads to

$$
\begin{aligned}
a(x,y) = &\frac{1}{2\pi} \int_0^{2\pi} \sum_{\substack{\nu,j \\ \nu',j'}} \Re\left(\kappa_{\nu,j}^*\right) \Re\left(\kappa_{\nu',j'}^*\right) \left|\varphi_{\nu,j}^{(x,y)} \varphi_{\nu',j'}^{(x,y)}\right| \cos\left(\theta_{\nu,j} - \theta_{\nu',j'} - \alpha\left(\nu - \nu'\right)\right) d\alpha \\
&+\frac{1}{2\pi} \int_0^{2\pi} \sum_{\substack{\nu,j \\ \nu',j'}} \Im\left(\kappa_{\nu,j}^*\right) \Im\left(\kappa_{\nu',j'}^*\right) \left|\varphi_{\nu,j}^{(x,y)} \varphi_{\nu',j'}^{(x,y)}\right| \cos\left(\theta_{\nu,j} - \theta_{\nu',j'} - \alpha\left(\nu - \nu'\right)\right) d\alpha \\
&-\frac{1}{\pi} \int_0^{2\pi} \sum_{\substack{\nu,j \\ \nu',j'}} \Im\left(\kappa_{\nu,j}^*\right) \Re\left(\kappa_{\nu',j'}^*\right) \left|\varphi_{\nu,j}^{(x,y)} \varphi_{\nu',j'}^{(x,y)}\right| \sin\left(\theta_{\nu,j} - \theta_{\nu',j'} - \alpha\left(\nu - \nu'\right)\right) d\alpha.
\end{aligned}
$$

In this equation, only the trigonometric functions are $\alpha$-dependent. Calculating the remaining integrals leads to

$$
\int_0^{2\pi} \mathrm{sc}\left(\theta_{\nu,j} - \theta_{\nu',j'} - \alpha\left(\nu - \nu'\right)\right) d\alpha = \begin{cases} 2\pi \; \mathrm{sc}\left(\theta_{\nu,j} - \theta_{\nu',j'}\right) & \textbf{if } \nu = \nu' \\ 0 \text{ otherwise,} \end{cases} \tag{5.4}
$$

where sc can represent the cosine or the sine function. Therefore,

$$
\begin{aligned}
a(x,y) = \sum_\nu \Bigg[ &\sum_{j,j'} \Re\left(\kappa_{\nu,j}^*\right) \Re\left(\kappa_{\nu,j'}^*\right) \left|\varphi_{\nu,j}^{(x,y)} \varphi_{\nu,j'}^{(x,y)}\right| \cos\left(\theta_{\nu,j} - \theta_{\nu,j'}\right) \\
&+\sum_{j,j'} \Im\left(\kappa_{\nu,j}^*\right) \Im\left(\kappa_{\nu,j'}^*\right) \left|\varphi_{\nu,j}^{(x,y)} \varphi_{\nu,j'}^{(x,y)}\right| \cos\left(\theta_{\nu,j} - \theta_{\nu,j'}\right) \\
&-2\sum_{j,j'} \Im\left(\kappa_{\nu,j}^*\right) \Re\left(\kappa_{\nu,j'}^*\right) \left|\varphi_{\nu,j}^{(x,y)} \varphi_{\nu,j'}^{(x,y)}\right| \sin\left(\theta_{\nu,j} - \theta_{\nu,j'}\right) \Bigg],
\end{aligned} \tag{5.5}
$$

and by using once again Equation (5.3), Equation (5.2) finally leads to

$$
a(x,y) = \sum_\nu \left| \sum_j \kappa_{\nu,j}^* \varphi_{\nu,j}^{(x,y)} \right|^2. \tag{5.6}
$$

Thanks to the use of Bessel coefficients instead of raw pixel values, the classic convolution has been modified into Equation (5.6) in order to achieve rotation equivariance. Nevertheless, by introducing reduction mechanisms in the models to make the feature maps in the final layer of size $1 \times 1$ (by using pooling layers or avoiding padding), the global equivariance can lead to global invariance. Figure 8 presents an example where the equivariance of B-CNNs is compared to vanilla CNNs, and it also presents how a succession of equivariant feature maps leads in this case to a global invariance of the model.

Finally, one can point out that $a(x,y) \in \mathbb{R}$ (even if $\kappa_{\nu,j} \in \mathbb{C}$ and $\varphi_{\nu,j}^{(x,y)} \in \mathbb{C}$). This is an important property as it allows this operation to be compatible with existing deep learning frameworks (for example, classic activation functions and batch normalization can be used), as opposed to the work of Worrall et al. (2017) that uses values in the complex domain. It is also worth to mention that this operation is *pseudo*-injective, meaning that different images will lead to different values of $a$ (*pseudo* makes reference to the exception when an image is compared to a rotated version of itself). The proof for the pseudo-injectivity is presented in Appendix C.

## 5.2 Adding the Reflection Equivariance

In order to make B-CNNs also equivariant to reflections, and thus $O(2)$ equivariant, one can check how Equation (5.6) behaves for an image and its reflection. To do so, let us compute the quantity

$$
\delta = \sum_\nu \left| \sum_j \kappa_{\nu,j}^* \varphi_{\nu,j} \right|^2 - \sum_\nu \left| \sum_j \kappa_{\nu,j}^* \varphi_{\nu,j}^{\mathrm{ref}} \right|^2,
$$

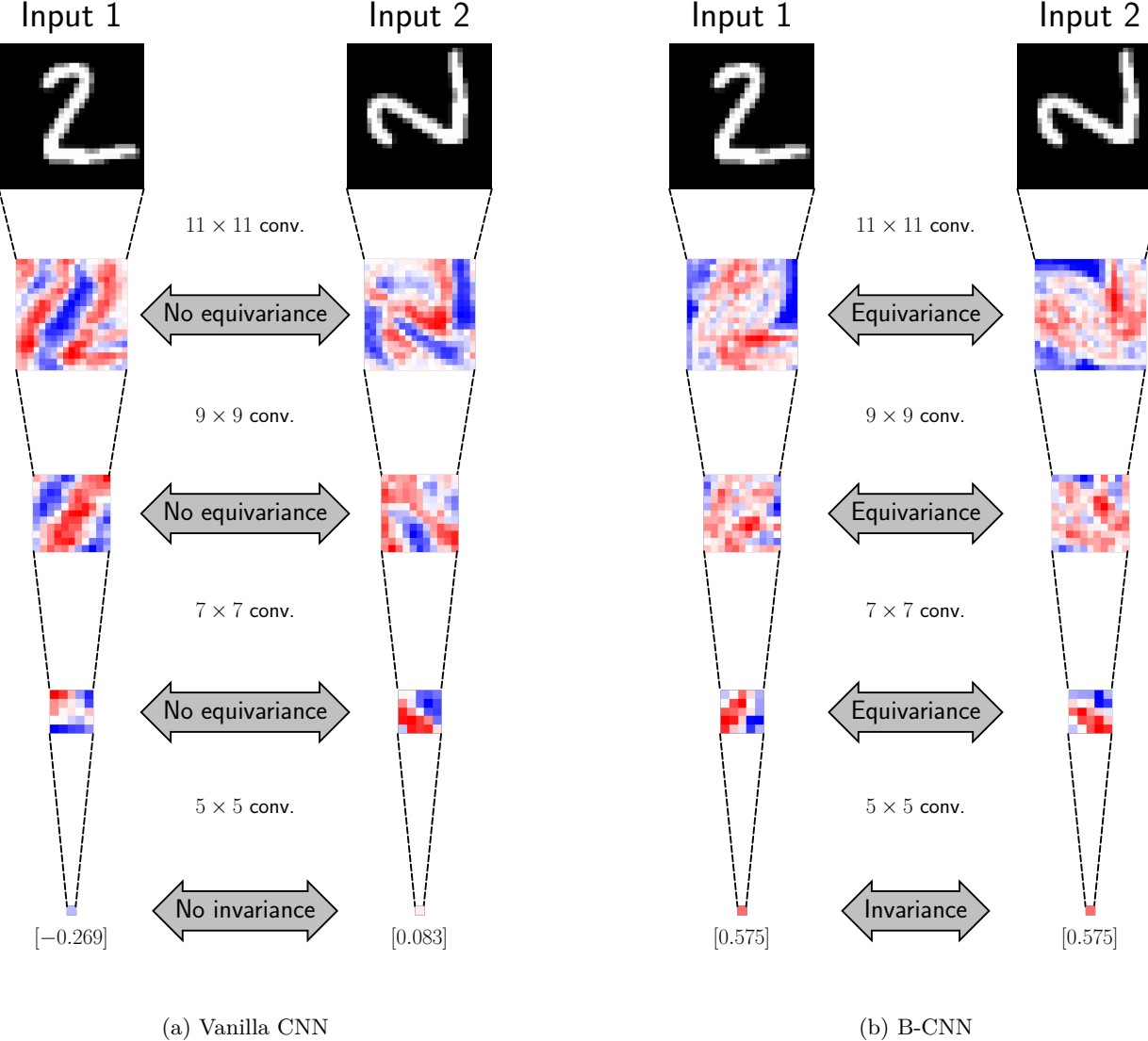

(a) Vanilla CNN                    (b) B-CNN

Figure 8: Feature maps obtained with 4 vanilla (a) and Bessel (b) convolutional layers on a sample drown from the MNIST data set (LeCun et al., 1998). Input 2 is the exact same image as Input 1, except for a $\frac{\pi}{2}$ rotation. Inputs are of size $29 \times 29$, and no padding is used such that the size of feature maps is progressively reduced until it reaches the final size of $1 \times 1$. One can observe that no equivariance is obtained in the feature maps for the vanilla CNN, leading to different final results. However, B-CNN provides equivariance for the feature maps, leading to a final result that is invariant to the orientation of the image. Indeed, for B-CNN, feature maps for Input 2 are rigorously identical up to a $\frac{\pi}{2}$ rotation, which is not the case for vanilla CNN. It has to be noted that in this example, equivariance/invariance are rigorously achieved because $\frac{\pi}{2}$ rotations are well-defined on Cartesian grids.

where $\varphi_{\nu,j}^{\text{ref}}$ are the Bessel coefficients of a reflected version of $\Psi(\rho, \theta)$. For the operation to be invariant under reflection, $\delta$ should therefore be equal to 0. Thanks to Equation (4.9) and Equation (4.6), we can write

$$
\begin{aligned}
\delta &= \sum_{\nu} \big| \sum_{j} \kappa_{\nu,j}^* \varphi_{\nu,j} \big|^2 - \sum_{\nu} \big| \sum_{j} \kappa_{\nu,j}^* \left( (-1)^{\nu} \Re(\varphi_{\nu,j}) + i (-1)^{\nu+1} \Im(\varphi_{\nu,j}) \right) \big|^2 \\
&= \sum_{\nu} \big| \sum_{j} \kappa_{\nu,j}^* \varphi_{\nu,j} \big|^2 - \sum_{\nu} \big| (-1)^{\nu} \sum_{j} \kappa_{\nu,j}^* \varphi_{\nu,j}^* \big|^2 \\
&= \sum_{\nu} \left[ \big| \sum_{j} \kappa_{\nu,j}^* \varphi_{\nu,j} \big|^2 - \big| \sum_{j} \kappa_{\nu,j}^* \varphi_{\nu,j}^* \big|^2 \right].
\end{aligned}
$$

By using again the development that led to Equation (5.5), one can show that

$$
\delta = -4 \sum_{\nu,j,j'} \Im\left( \kappa_{\nu,j}^* \right) \Re\left( \kappa_{\nu,j'}^* \right) \left| \varphi_{\nu,j} \varphi_{\nu,j'} \right| \sin\left( \theta_{\nu,j} - \theta_{\nu,j'} \right).
$$

It means that $\delta$ may be different from 0 and an $O(2)$ equivariance will in general not be achieved. The objective is now to slightly modify Equation (5.6) in order to obtain $\delta = 0$. To do so, one can see that the terms that do not vanish are those that involve $\Im\left( \kappa_{\nu,j}^* \right) \Re\left( \kappa_{\nu,j'}^* \right)$. By avoiding such crossed terms between the real and imaginary parts of $\kappa_{\nu,j}$, one can obtain $\delta = 0$ and therefore a reflection equivariance, while still keeping the rotation equivariance. This can be achieved by using

$$
a(x,y) = \sum_{\nu} \big| \sum_{j} \Re\left( \kappa_{\nu,j}^* \right) \varphi_{\nu,j}^{(x,y)} \big|^2 + \big| \sum_{j} \Im\left( \kappa_{\nu,j}^* \right) \varphi_{\nu,j}^{(x,y)} \big|^2. \tag{5.7}
$$

To conclude, B-CNNs can be made $SO(2)$ equivariant (that is, equivariant to all the continuous planar rotations) by using Equation (5.6) as operation between the filters and the images, or $O(2)$ equivariant (that is, equivariant to all the continuous planar rotations and reflections) by using Equation (5.7) instead. Users can decide, based on the application, which equivariance is required.

## 6 Bessel-Convolutional Neural Networks

This section constitutes a sum up and gives more intuition about the global working of B-CNNs. It also presents an efficient way for implementing the previous developments in convolutional neural networks architectures. It is finally shown how bringing multi-scale equivariance is straightforward with this implementation. Multi-scale equivariance means that patterns can be detected even if they appear at slightly different scales in the images. Developments in this section are presented in the particular case of $SO(2)$ equivariance. We will hence consider Equation (5.6) instead of Equation (5.7). However, developments can easily be adapted for this second case.

### 6.1 B-CNNs From a Practical Point of View

The key modification in B-CNNs compared to vanilla CNNs is to replace the element-wise multiplication between raw pixel values and the filters by the mathematical operation described by Equation (5.6). Filters, which are described by their Bessel coefficients $\{\kappa_{\nu,j}\}$, sweep locally the image and the Bessel coefficients for the sub-region of the image $\{\varphi_{\nu,j}^{(x,y)}\}$ are computed. Equation (5.6) is then used to progressively build feature maps. This process is summarized in Figure 9. However, implementing B-CNNs by using this straightforward strategy requires to perform many Bessel coefficients decompositions, which are really expensive.

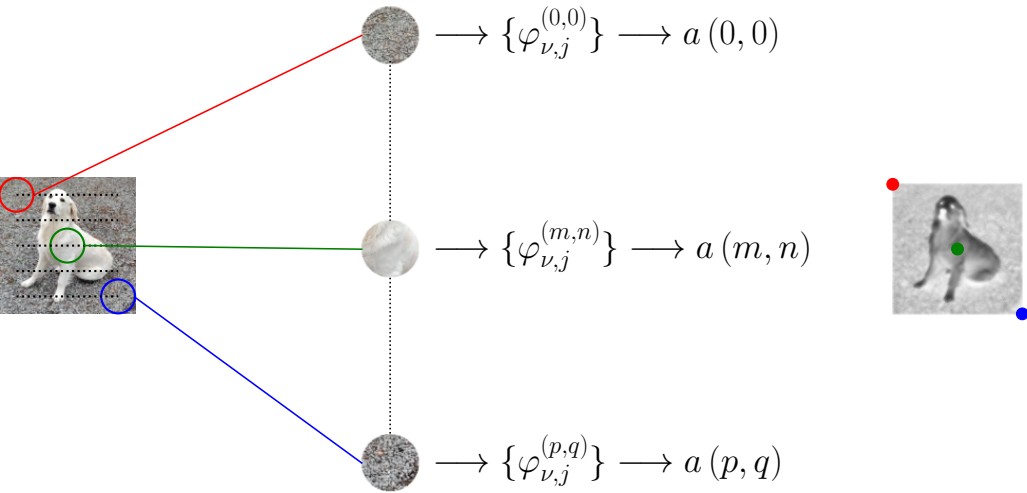

Figure 9: Illustration of how feature maps are obtained in B-CNNs. One can see that B-CNNs still work in a convolutional fashion, but the key operation between the filters and the image is modified.

A more efficient implementation can be obtained by developing Equation (5.6) with Equation (4.3). Indeed, it gives

$$a\left(x,y\right) = \sum_{\nu} |\sum_{j} \kappa_{\nu,j}^{*} \int_{0}^{2\pi} \int_{0}^{R} \rho \left[N_{\nu,j} J_{\nu}\left(k_{\nu,j}\rho\right) e^{i\nu\theta}\right]^{*} \Psi^{(x,y)}\left(\rho,\theta\right) d\rho d\theta|^{2}$$

$$= \sum_{\nu} |\int_{0}^{2\pi} \int_{0}^{R} \Psi^{(x,y)}\left(\rho,\theta\right) \sum_{j} \rho \left[N_{\nu,j} J_{\nu}\left(k_{\nu,j}\rho\right) e^{i\nu\theta}\right]^{*} \kappa_{\nu,j}^{*} d\rho d\theta|^{2},$$

and, by converting to Cartesian coordinates thanks to $\widetilde{\theta} \equiv \widetilde{\theta}\left(x,y\right) = \arctan\frac{y}{x}$ and $\widetilde{\rho} \equiv \widetilde{\rho}\left(x,y\right) = \sqrt{x^2 + y^2}$, it leads to

$$a\left(x,y\right) = \sum_{\nu} |\int_{-R}^{R} \int_{-R}^{R} \Psi^{(x,y)}\left(x',y'\right) \sum_{j} \left[N_{\nu,j} \widetilde{J}_{\nu}\left(k_{\nu,j}\widetilde{\rho}\right) e^{i\nu\widetilde{\theta}}\right]^{*} \kappa_{\nu,j}^{*} dx'dy'|^{2}, \tag{6.1}$$

where $\widetilde{J}_{\nu}\left(k_{\nu,j}\rho\right)$ is defined as

$$\widetilde{J}_{\nu}\left(k_{\nu,j}\rho\right) = \begin{cases} J_{\nu}\left(k_{\nu,j}\rho\right) \text{ if } \rho \leq R \\ 0 \text{ otherwise.} \end{cases}$$

This definition of $\widetilde{J}_{\nu}$ is required to compensate for the fact that we are now integrating over the square domain $[-R, R] \times [-R, R]$ instead of the circular domain $D^2$ of radius $R$. By defining

$$T_{\nu,j}\left(x,y\right) = N_{\nu,j} \widetilde{J}_{\nu}\left(k_{\nu,j}\widetilde{\rho}\right) e^{-i\nu\widetilde{\theta}}, \tag{6.2}$$

one can finally obtain

$$a\left(x,y\right) = \sum_{\nu} \left|\Psi\left(x,y\right) * \sum_{j} T_{\nu,j}\left(x,y\right) \kappa_{\nu,j}^{*}\right|^{2}$$

$$= \sum_{\nu} \left|\Psi\left(x,y\right) * F_{\nu}\left(x,y\right)\right|^{2}, \tag{6.3}$$

where

$$F_\nu(x,y) = \sum_j T_{\nu,j}(x,y)\,\kappa_{\nu,j}^*. \tag{6.4}$$

From a numerical point of view, there are two main advantages in using Equation (6.3) instead of directly implementing Equation (5.6) as presented in Figure 9. Firstly, this equation directly involves the input $\Psi(x,y)$ instead of its Bessel coefficients. Secondly, $T_{\nu,j}(x,y)$ does not depend on the input or the weights of the model. Therefore, it can be computed only once at the initialization of the model. After discretizing space, $T_{\nu,j}(x,y)$ can be seen as a transformation matrix that maps the weights of the model from the Fourier-Bessel transform domain (the Bessel coefficients of the filters $\{\kappa_{\nu,j}\}$) to a set of filters in the direct space $\{F_\nu(x,y)\}$. The feature maps can then be obtained by applying classic convolutions between the input and these filters. Also note that the $\nu_{\max}$ convolutions that need to be performed can be wrapped with the output channel dimension to only perform one call to the convolution function. Algorithm 1 presents how to efficiently implement a B-CNN layer in practice, including the initialization step and the forward propagation.

---

**Algorithm 1:** Implementation of a B-CNN layer

---

`/* Initialization (performed only one time)` `*/`

**Input:** The size of the filters $(2n+1)$; the number of output channels $C_{out}$

Compute $k_{\max} = \frac{2n+1}{2}\pi$, $\nu_{\max}$ and $j_{\max}$ (Equation 4.7)

Let $\mathbf{K}$ be a randomly-initialized trainable tensor with shape $[\nu_{\max}, j_{\max}, C_{in} \times C_{out}]$
**forall** $(\nu, j)$ *s.t.* $k_{\nu,j} > k_{\max}$ **do**
    `/* Parameters s.t. ` $k_{\nu,j} > k_{\max}$ `are set to 0` `*/`
    $\mathbf{K}[\nu, j, :] = 0$

Let $\mathbf{T}$ be a zero-initialized tensor with shape $[\nu_{\max}, 2n+1, 2n+1, j_{\max}]$
**forall** $(\nu, j)$ *s.t.* $k_{\nu,j} \leq k_{\max}$ **do**
    **forall** $(x,y) \in \{-1, -1+\frac{1}{n}, \ldots, 1-\frac{1}{n}, 1\} \times \{-1, -1+\frac{1}{n}, \ldots, 1-\frac{1}{n}, 1\}$ **do**
        $\mathbf{T}[\nu, x, y, j] = N_{\nu,j}\widetilde{J}_\nu\left(k_{\nu,j}\sqrt{x^2+y^2}\right)e^{-i\nu\left(\arctan\frac{y}{x}\right)}$ (Equation 6.2)

Reshape $\mathbf{T}$ into $[\nu_{\max}, (2n+1) \times (2n+1), j_{\max}]$

---

`/* Forward propagation` `*/`

**Input** : A tensor $\mathbf{I}$ of $N$ images of size $[N, W, H, C_{in}]$
**Output:** The tensor $\mathbf{A}$ with the corresponding feature maps

Let $\mathbf{F}$ be a tensor with shape $[\nu_{\max}, (2n+1) \times (2n+1), C_{in} \times C_{out}]$
**for** *(*$\nu = 0$*;* $\nu < \nu_{\max}$*;* $\nu = \nu + 1$*)* **do**
    `/* ` $\mathbf{F}$ ` can be computed thanks to ` $\nu_{\max}$ ` matrix multiplications` `*/`
    $\mathbf{F}[\nu, :, :] = \mathrm{matmul}\left(\mathbf{T}[\nu, :, :], \mathbf{K}[\nu, :, :]\right)$ (Equation 6.4)
Reshape $\mathbf{F}$ into $[(2n+1), (2n+1), C_{in}, \nu_{\max}C_{out}]$

`/* ` $W_{out}$ ` and ` $H_{out}$ ` depend on the padding and the stride` `*/`
$\mathbf{Z} = \mathrm{conv2d}(\mathbf{I}, \mathbf{F})$
Reshape $\mathbf{Z}$ into $[N, W_{out}, H_{out}, C_{out}, \nu_{\max}]$

Let $\mathbf{A}$ be a zero-initialized tensor with shape $[N, W_{out}, H_{out}, C_{out}]$
**for** *(*$\nu = 0$*;* $\nu < \nu_{\max}$*;* $\nu = \nu + 1$*)* **do**
    $\mathbf{A} \mathrel{+}= \left|\mathbf{Z}[:, :, :, :, \nu]\right|^2$ (Equation 6.3)
**return $\mathbf{A}$**

---

### 6.2 Numerical Complexity of B-CNNs

Regarding the computational complexity, if the input of a vanilla CNN layer is of size $[W \times H \times C_{in}]$ and if it implements $C_{out}$ filters of size $[2n \times 2n]$, the number of mathematical operations to perform for a forward pass (assuming that padding is used along with unitary strides) is

$$N_{op} = WH \left[ 4n^2 C_{in} + \left( 4n^2 C_{in} - 1 \right) \right] C_{out}$$
$$= 8WHn^2 C_{in} C_{out} - WHC_{out}.$$

Indeed, $C_{out}$ filters made of $2n \times 2n \times C_{in}$ parameters will sweep $WH$ local parts of the input image. For each local part, $4n^2 C_{in}$ multiplications are then performed as well as $4n^2 C_{in} - 1$ additions. Therefore, it leads to a computational complexity of $\mathcal{O}\left( WHn^2 C_{in} C_{out} \right)$. Compared to vanilla CNNs, B-CNNs need to perform more operations as it is required (i) to compute $F_\nu(x, y)$, (ii) to perform $2\nu_{\max}$ times more convolutions and (iii) to compute squared modulus and a sum over $\nu$. Step (i) consists of $\nu_{\max}$ matrix multiplications, which involve for each element in the final matrix $j_{\max}$ scalar multiplications and $j_{\max} - 1$ additions. Step (ii) is the same that for vanilla CNNs, except that one should perform this for each $\nu$ and both for the real and imaginary parts of $F_\nu(x, y)$. Finally, the squared modulus in step (iii) involves $2WHC_{out}$ multiplications and $WHC_{out}$ additions, and the final summation over $\nu$ involves $\nu_{\max} - 1$ additions. At the end, the final numbers of operations to perform for each step are

$$N_{op}^{(i)} = \nu_{\max} \left[ j_{\max} + (j_{\max} - 1) \right] 4n^2 C_{in} C_{out},$$
$$N_{op}^{(ii)} = WH \left[ 4n^2 C_{in} + \left( 4n^2 C_{in} - 1 \right) \right] 2C_{out} \nu_{\max},$$
$$N_{op}^{(iii)} = 3WHC_{out} \nu_{\max} + (\nu_{\max} - 1) WHC_{out}.$$

However, by looking at Figure 10, one can see that both $\nu_{\max}$ and $j_{\max}$ scale linearly with $n$, thanks to the constraint expressed by Equation (4.7). It follows that

$$N_{op} \propto n^4 C_{in} C_{out} + WHn^3 C_{in} C_{out} + WHn C_{out},$$

resulting in a computational complexity of $\mathcal{O}\left( WHn^3 C_{in} C_{out} \right)$ for a forward pass in a B-CNN layer. Since generally $n \ll \min\left( W, H, C_{in}, C_{out} \right)$, the increase in computational time compared to vanilla CNNs is reasonable with respect to the positive impact of the equivariance. Furthermore, the computational complexity of $E(2)$-equivariant models (Weiler & Cesa, 2019) for a symmetry group $\mathcal{G}$ is $\mathcal{O}\left( WHn^2 C_{in} C_{out} |\mathcal{G}| \right)$ as $C_{out}$ is artificially increased by the number of discrete operations in $\mathcal{G}$. Therefore, if $n < |\mathcal{G}|$ (which is generally the case as $n = 5, 7$ or $9$, and $\mathcal{G} = C_8, C_{16}, D_8$ or $D_{16}$ leading to $|\mathcal{G}| = 8, 16, 16$ or $32$, respectively), B-CNNs are therefore more efficient from a computational point of view.

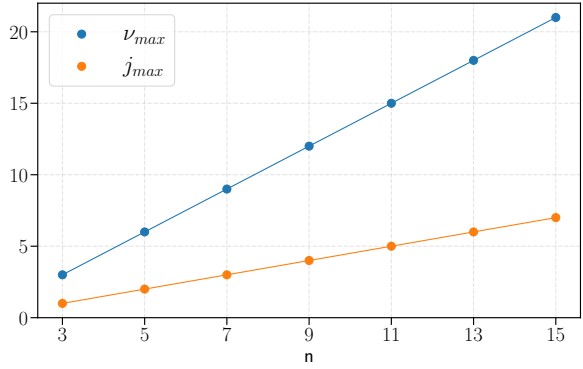

Figure 10: Relation between $n$ and $\nu_{\max}/j_{\max}$ thanks to Equation (4.7).

### 6.3 Rotation Equivariance From a Numerical Point of View

As opposed to most of the state-of-the-art methods, B-CNNs do not rely on a particular discretization of the continuous $(S\text{-})O(2)$ group. The equivariance is automatically guaranteed by processing the input image thanks to an $(S\text{-})O(2)$ equivariant mathematical operation, which replaces the simple convolution in the direct space. It follows that B-CNNs directly provide theoretical guarantees regarding the equivariance to the continuous set of rotation angles $[0, 2\pi[$. Indeed, as the Bessel coefficients of the filter $\{\kappa_{\nu,j}\}$ are not computed but defined as the learnable parameters of the model, it does not involve any numerical error. Furthermore, Equation (5.6) is rotation invariant regardless of the number of Bessel coefficients used (that is, independently of $k_{\max}$). However, one should mention that, from a numerical point of view, exact $(S\text{-})O(2)$ equivariance is rarely possible due to the discrete nature of numerical images. Indeed, $\Psi(x, y)$ is only known on a finite Cartesian grid, and rotations of angles in $[0, 2\pi[ \, \setminus \, \{0, \frac{\pi}{2}, \pi, \frac{3\pi}{2}\}$ are not well defined, and will result in numerical errors. The only source of errors in B-CNNs regarding the $(S\text{-})O(2)$ equivariance lies in the discretization of $\Psi^{(x,y)}(x', y')$ on an $[2n \times 2n]$ Cartesian grid, which is involved by Equation (6.1). Therefore, numerical errors may be reduced by increasing $n$, that is, the size of the filters.

### 6.4 Adding a Multi-Scale Equivariance

Previous sections focus on achieving $SO(2)$ and $O(2)$ equivariance. However, for particular applications, patterns of interest may also vary in scale. To illustrate this, see for example biomedical applications where tumors may be of different sizes. Prior works (Xu et al., 2014; Li et al., 2019; Ghosh & Gupta, 2019) already show that a multi-scale equivariance can be incorporated into CNNs, leading to better performances for such applications. The aim of this section is to present how these prior works can be easily transposed to the particular case of B-CNNs.

As the size of the filter in the direct space is determined by the discretization of $T_{\nu,j}(x, y)$, it is easy in B-CNNs to implement already-existing scaling invariance techniques. To do so, we only need to pre-compute multiple versions of $T_{\nu,j}(x, y)$ for different kernel sizes, and only keep the one with the highest response. More formally, the idea is to define multiple transformation matrices $T_{\nu,j}^n(x, y)$ that act on circular domains of different size $n$. Those matrices can be pre-computed at initialization. They can then be used to project the filters in the Fourier-Bessel transform domain to filters of different sizes in the direct space. One can then consider keeping only the most active feature maps. The process is summarized in Figure 11.

## 7 Experiments

This section presents the details of all the experiments performed to assess and to compare the equivariance obtained with B-CNNs with other state-of-the-art methods. The data sets used are presented, as well as the experimental setup. After that, quantitative results are presented for each data set.

### 7.1 Data sets

Three data sets are used to assess the performances in different practical situations:

- The MNIST (LeCun et al., 1998) data set is a classical baseline for image classification. This data set is made of $28 \times 28$ grayscale images of handwritten digits that belong therefore to one out of 10 different classes. More precisely, four variants of this data set are considered: (i) MNIST, (ii) MNIST-rot, (iii) MNIST-back and (iv) MNIST-rot-back[4]. In the *rot* variants, images are randomly rotated by an angle $\alpha \in [0, 2\pi[$. In the *back* variants, a patch from a black and white image was used as the background for the digit image. This adds useless information that can be disturbing for some architectures. All these MNIST data sets are perfectly balanced.

- The Galaxy10 DECals data set is a subset of the original Galaxy Zoo data set (Willett et al., 2013). This data set is initially made of $256 \times 256$ RGB images of galaxies that belong to one out of 10

---

[4]All these variants are generated from the initial MNIST data set, and can be found at `https://sites.google.com/a/lisa.iro.umontreal.ca/public_static_twiki/variations-on-the-mnist-digits`

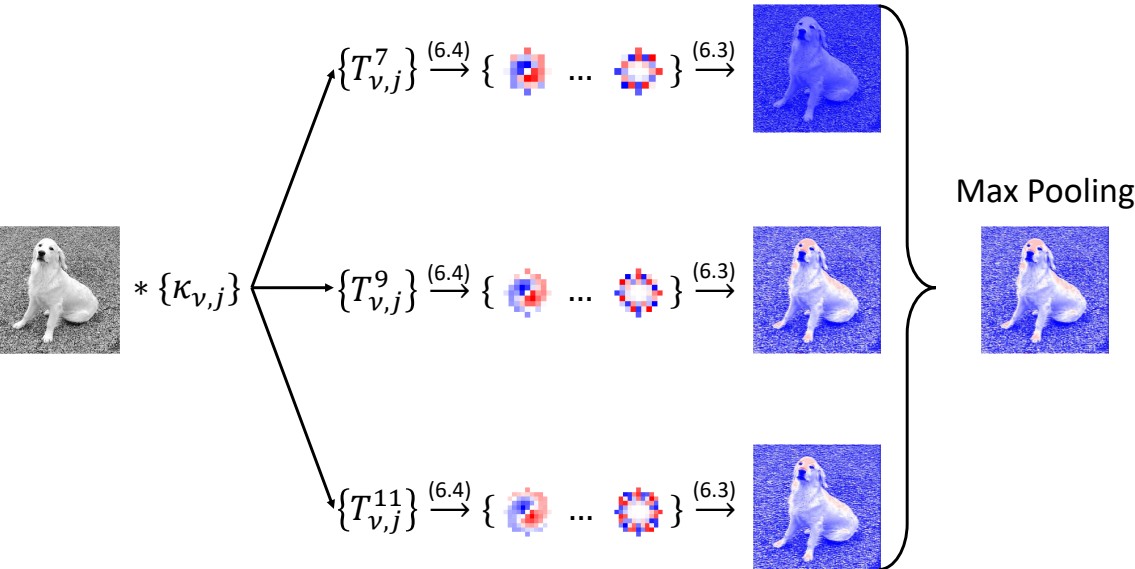

Figure 11: This figure presents how B-CNNs can handle multi-scale equivariance. In this case, a single filter represented by its Bessel coefficients $\{\kappa_{\nu,j}\}$ is projected in the direct space thanks to different transformation matrices. The filter is mapped to $\nu$ filters of size $7 \times 7$, $9 \times 9$ and $11 \times 11$. Max pooling is then used to only keep the most responding feature maps. Note that only the real parts of the projected filters are represented here, for convenience.

roughly balanced classes, representing different possible morphologies according to experts. Images are resized to $128 \times 128$ in our work for computational resources purpose.

- The Malaria (Yu et al., 2020) data set is made of $64 \times 64$ RGB microscope images of blood films. Those images belong to two perfectly balanced classes highlighting the presence or not of the parasites responsible for Malaria.

An overview for all those three data sets is presented in Table 1, along with visual examples.

## 7.2 Experimental Setup

In order to perform this empirical study, (i) $E(2)$-equivariant CNNs from Weiler & Cesa (2019) ($E(2)$-CNNs), (ii) Harmonic Networks from Worrall et al. (2017) (HNets) as well as (iii) vanilla CNNs are considered along with B-CNNs. This choice is motivated by the fact that $E(2)$-CNN and HNets constitute the state of the art for constraining CNNs with known symmetry groups. Each technique is tested in different setups (mainly, for different symmetry groups or different representations of the same group). The different setups for each method are described below:

- For $E(2)$-CNNs, we consider the discrete $C_4$ ($\{n\frac{\pi}{2}\}_{n=1}^4$ rotations), $C_8$ ($\{n\frac{\pi}{4}\}_{n=1}^8$ rotations) and $C_{16}$ ($\{n\frac{\pi}{8}\}_{n=1}^{16}$ rotations) symmetry groups using a regular representation, as well as the continuous one, $SO(2)$ (all the continuous rotations) and $O(2)$ (all the continuous rotations and the reflections along vertical and horizontal axes), using irreducible representations. Those setups are a subset of all the setups tested by the authors of $E(2)$-CNNs. More details about this and how $E(2)$-CNNs work can be found in the work of Weiler & Cesa (2019). Furthermore, the authors provide an implementation for $E(2)$-CNN that has been used in this work.

Table 1: Overview of the data sets. $N$ is the total number of available data, and $C$ is the number of classes.

| Data set | $N$ | $C$ | Task | Resolution | Examples |
|---|---|---|---|---|---|
| MNIST

-rot

-back

-rot-back | $62,000$ | $10$ | Multi-class classification | $28 \times 28 \times 1$ |  |
| Galaxy10 DECals | $17,736$ | $10$ | Multi-class classification | $128 \times 128 \times 3$ |  |
| Malaria | $27,558$ | $2$ | Binary classification | $64 \times 64 \times 3$ |  |

- For HNets, similarly to what the authors did in their work, two different setups to achieve $SO(2)$ invariance are tested using an approximation to the first and second order. In this work, we use again the implementation provided by the authors of $E(2)$-CNNs, who re-implement HNets in their own framework, for convenience.

- Regarding B-CNNs, four setups are considered to achieve $SO(2)$ or $O(2)$, with or without scale invariance (denoted by the presence or not of "+" in our tables and figures), with the computation of $k_{\max}$ as described by Equation (4.7). Another setup for $SO(2)$ invariance with a lower cutoff frequency, which corresponds to half the initial $k_{\max}$ is also considered. This last setup is motivated by the empirical observation that it often leads to better performances. Indeed, removing some meaningless high frequency information sometimes have a positive impact on the networks. There is also a similar idea in the work of Weiler & Cesa (2019) for $E(2)$-CNNs, by applying a Gaussian blur when pooling the images. This idea was therefore also used as presented in their work for $E(2)$-CNNs.

- Finally, a vanilla CNN with the same architecture as for the other methods, as well as a ResNet-18 (He et al., 2016) are also trained for reference.

The architectures are strongly inspired from the work of Weiler & Cesa (2019) and are presented in a generic fashion in Table 2. Note that the size of the filters is larger than conventional sizes in CNNs. This is also the case in the related works, and a justification for this is presented in Section 6.3. The same template architecture is used for all the methods (except for the ResNet-18 architecture that is kept unmodified) and data sets. Nonetheless, minor modifications are sometimes performed. Firstly, the number of filters in each convolutional layer should be adapted from one method to another, in order to keep the same

number of trainable parameters. To do so, a parameter $\lambda$ is introduced to manually scale the number of filters and guarantee the same number of trainable parameters for all the methods. To give an idea, $\lambda$ is arbitrarily set to 1 for B-CNNs with the high cutoff frequency policy ($k_{\max}$), and the corresponding number of trainable parameters is close to $115,000$. Secondly, $E(2)$-CNNs require a particular operation called *invariant projection* before applying the dense layer. This specific operation is not performed for the other methods. Thirdly, each convolutional layer is followed by a batch-normalization and a *ReLU* activation function, except for B-CNNs. Indeed, we empirically observed that both the batch-normalization and the *ReLU* activation function generally decrease convergence for B-CNNs, while this is not the case for the other methods. Therefore, we do not consider any batch-normalization layer for B-CNNs and we use *Tanh* activation functions, which seems to perform better in our case. Note that we are still not able to really understand why classic batch-normalization and *ReLU* activation functions reduce the performances of B-CNNs. Yet, we clearly observed empirically that using batch-normalization makes the convergence of B-CNNs much slower, and using *ReLU* can hurt the optimization process to a point where the model is only marginally better than random predictions. Regarding the *ReLU* activation function, a possibility could be the fact that B-CNNs involve a real and positive representation (due to the use of a square modulus). Therefore, using a *ReLU* activation function is equivalent to not using any activation function (as *ReLU* is linear for $x >= 0$). Nonetheless, the introduction of a bias makes it theoretically possible to arbitrarily shift the representation to negative values, and the real reason why *ReLU* is not working with B-CNNs is still partially unknown. Finally, the padding and the final layer are adapted according to the considered data set, as they involve different tasks and image sizes.

Table 2: Generic architecture used for the different data sets. *Conv layer* can either be vanilla-conv, B-conv, $E(2)$-conv or HNet-conv. After each *Conv layer*, a batch-normalization as well as an activation function are applied (*Tanh* for B-conv, *ReLU* for others), except for B-conv that does not use any batch-normalization. The *Invariant projection* is only required in $E(2)$-CNNs. As the number of parameters in a filter may differ between the different methods, a parameter $\lambda$ is introduced. This parameter is fixed for each method in order to tweak the number of filters (# C) so that the total numbers of trainable parameters are as close as possible to each others.

| Layer | # C | MNIST(-rot)(-back) | Galaxy10 DECals | Malaria |
|---|---|---|---|---|
| *Conv layer* $9 \times 9$ | $8\lambda$ | pad 4 | pad 0 | pad 4 |
| *Conv layer* $7 \times 7$ | $16\lambda$ | pad 3 | pad 0 | pad 3 |
| Av. pool. $2 \times 2$ | - | pad 0 | pad 0 | pad 0 |
| | | | | |
| *Conv layer* $7 \times 7$ | $24\lambda$ | pad 3 | pad 0 | pad 3 |
| *Conv layer* $7 \times 7$ | $24\lambda$ | pad 3 | pad 0 | pad 0 |
| Av. pool. $2 \times 2$ | - | pad 0 | pad 0 | pad 0 |
| | | | | |
| *Conv layer* $7 \times 7$ | $32\lambda$ | pad 3 | pad 0 | pad 0 |
| *Conv layer* $7 \times 7$ | $40\lambda$ | pad 0 | pad 0 | pad 0 |
| (*Inv. projection*) | - | - | - | - |
| | | | | |
| Global av. pool. | - | - | - | - |
| Dense layer | $\rightarrow$ | 10, *softmax* | 10, *softmax* | 2, *softmax* |

As it is expected that constraining CNNs with symmetry groups becomes more useful when less data are available (as CNNs should no more learn the invariances by themselves), experiments are performed in (i) High, (ii) Intermediate and (iii) Low data settings for each data set, while considering the use of data augmentation strategies or not. A last setup is also considered for the MNIST variants for which models are trained on the original MNIST data sets (no rotations are provided during training) and tested on rotated versions of the test images. The different data settings correspond to different sizes for the training sets. Attention is paid to keep the same percentage of samples of each target class, in order to avoid biases.

For the MNIST data sets, those settings correspond to the use of (i) 20%, (ii) 2% and (iii) 0.2% of the total number of available data for training. On top of this, three different data augmentation policies are tested.

Firstly, models are trained on MNIST-rot(-back) using online data augmentation (by performing random rotation before being given as input). Secondly, models are still trained on MNIST-rot(-back) but without further data augmentation (the same image is always seen by the model in the same orientation). Thirdly, models are trained on MNIST(-back) while being tested on random rotated versions of the test images. Those setups allow us to see how the amount of data impact the performance of the models, and how much the models still rely on the training phase to achieve the desired invariances.

For the other data sets, the different data settings correspond to the use of (i) 80%, (ii) 8% and (iii) 0.8% of the total number of available training data, respectively. As for the MNIST data sets, models are again trained with and without using data augmentation. However, in this case, the data augmentation also performs random planar reflections (not pertinent for MNIST). Also note that only two data augmentation policies are possible, because it is meaningless to think of non-rotated version of images for Galaxy10 DECals and Malaria (as opposed to MNIST where digits have a well-defined orientation, a priori). In other words, the last setup cannot be considered for those last two data sets.

For the High and Intermediate data setting experiments, models are trained using the Adam optimizer through 50 epochs. A warm-up cosine decay scheduler that progressively increases the learning rate from 0 to 0.001 during the first 10 epochs before slowly decreasing it to 0 following a cosine function during the remaining epochs is used. For the low data setting experiments, 150 epochs are performed, with the warm-up phase during the first 30 epochs. Each experiment is performed on the exact same hardware (using Nvidia A100, 40 GB) through 5 independent runs. To ensure fair comparison of computational time, all the implementations use the same PyTorch version.

## 7.3 Results on MNIST(-rot)

Table 3 presents the results obtained on the MNIST(-rot) data sets. For the sake of completeness, Figure 12 and 13 also present all the corresponding training curves with respect to the epoch and the wall time, respectively.

From a general point of view, one can observe that a proper use of $E(2)$-CNNs, HNets and B-CNNs can lead to better performances than vanilla CNNs, even if the number of parameters is much smaller in the case of equivariant models ($\pm 115,000$ parameters against $\pm 11,000,000$ for ResNet-18). Vanilla CNNs techniques are only able to compete with equivariant models in high data settings, and when performing data augmentation (first column). This clearly highlights the fact that vanilla CNNs are sensitive to the quality and the amount of data in order to learn the invariances. Furthermore, even in the most favorable situation for vanilla CNNs, convergence is much slower than for equivariant models.

Next, by taking a closer look at the equivariant models, it appears that the $E(2)$-CNNs that use the straightforward discrete groups $C_4$, $C_8$ or $C_{16}$ perform quite well, and are even the best performing models when used along with data augmentation (first 3 columns). However, performances fall a little on the MNIST-rot data set without data augmentation (middle 3 columns), and becomes really bad compared to the $(S-)O(2)$ equivariant models when trained on the MNIST data set, when they cannot see rotated versions of the digits (last 3 columns). Even if those models are better than vanilla CNNs, it also appears that they still rely on training to learn really continuous rotation invariance, which is something expected. It is also interesting to mention that using a symmetry group that is not appropriate may be worse than not using any symmetry group at all. For example, in high data setting with data augmentation, vanilla CNNs perform better than $O(2)$-based models.

Interestingly, almost all the $(S-)O(2)$ equivariant models seem to achieve very similar performances on MNIST-rot, with and without data augmentation. Nonetheless, one can observe that the $SO(2)$ B-CNNs with the low cutoff policy ($k_{\max}/2$) is most of the time in the top-3 performing models, and achieve significantly better results than all the other models when only trained on MNIST (last 3 columns). This highlights the fact that the $SO(2)$ invariance achieved by design in the B-CNNs is stronger than the one achieved by other models, allowing generalization to rotated versions of digits, even if none of those are observed during training.

Table 3: Classification accuracy obtained for the different methods on the MNIST-rot and MNIST data sets, with and without data augmentation. For the last columns (MNIST, no rotation during training), models are trained on the standard MNIST data set and tested on rotated versions of digits to assess how models can generalize to orientations that are not observed during training. The *High*, *Inter.* and *Low* data regimes correspond to 12,000, 1,200 and 120 images for the training set (same percentage of samples of each target class), respectively. For each column, bold is used to highlight the top-3 performing models. Accuracy and standard deviation are assessed using 5 independent runs. The corresponding training curves are presented in Figure 12 and 13.

| Method | Group | MNIST-rot With data aug. High | Inter. | Low | MNIST-rot Without data aug. High | Inter. | Low | MNIST No rotation during training High | Inter. | Low |
|---|---|---|---|---|---|---|---|---|---|---|
| Vanilla CNN | {e} | $98.31 \pm 0.03$ | $94.70 \pm 0.13$ | $84.16 \pm 0.60$ | $95.87 \pm 0.18$ | $80.87 \pm 0.38$ | $44.45 \pm 2.44$ | $43.43 \pm 0.55$ | $37.40 \pm 1.87$ | $29.44 \pm 0.17$ |
| ResNet-18 | {e} | $98.66 \pm 0.05$ | $94.68 \pm 0.21$ | $80.16 \pm 0.20$ | $96.54 \pm 0.04$ | $76.19 \pm 9.01$ | $36.92 \pm 0.07$ | $43.24 \pm 0.16$ | $35.85 \pm 1.88$ | $28.29 \pm 1.62$ |
| *E(2)-CNN (regular)* | $C_4$ | $\mathbf{99.07 \pm 0.03}$ | $\mathbf{96.81 \pm 0.19}$ | $\mathbf{88.98 \pm 0.57}$ | $98.42 \pm 0.05$ | $94.18 \pm 0.12$ | $69.22 \pm 0.69$ | $68.94 \pm 1.43$ | $65.28 \pm 1.14$ | $56.45 \pm 1.98$ |
| | $C_8$ | $\mathbf{99.18 \pm 0.03}$ | $\mathbf{97.01 \pm 0.12}$ | $\mathbf{89.68 \pm 1.15}$ | $98.71 \pm 0.05$ | $94.60 \pm 0.30$ | $72.81 \pm 2.87$ | $68.52 \pm 0.74$ | $70.40 \pm 2.05$ | $62.19 \pm 2.74$ |
| | $C_{16}$ | $\mathbf{99.09 \pm 0.04}$ | $\mathbf{96.61 \pm 0.12}$ | $87.10 \pm 1.22$ | $98.53 \pm 0.06$ | $94.05 \pm 0.50$ | $62.46 \pm 5.53$ | $68.79 \pm 2.23$ | $69.34 \pm 4.28$ | $57.07 \pm 1.06$ |
| *E(2)-CNN (irr. $\leq 1$)* | $SO(2)$ | $98.86 \pm 0.04$ | $95.78 \pm 0.15$ | $84.71 \pm 0.24$ | $98.62 \pm 0.06$ | $95.30 \pm 0.34$ | $79.80 \pm 1.15$ | $\mathbf{97.66 \pm 0.17}$ | $94.12 \pm 0.30$ | $76.94 \pm 0.85$ |
| | $O(2)$ | $97.07 \pm 0.05$ | $90.39 \pm 0.09$ | $73.77 \pm 1.31$ | $98.30 \pm 0.02$ | $88.62 \pm 0.70$ | $68.95 \pm 1.73$ | $92.36 \pm 0.19$ | $86.21 \pm 0.23$ | $66.58 \pm 3.77$ |
| *E(2)-CNN (irr. $\leq 3$)* | $SO(2)$ | $98.76 \pm 0.07$ | $95.01 \pm 0.23$ | $80.14 \pm 0.96$ | $98.34 \pm 0.06$ | $90.98 \pm 0.50$ | $72.84 \pm 0.97$ | $96.11 \pm 0.09$ | $89.14 \pm 0.08$ | $68.69 \pm 1.50$ |
| | $O(2)$ | $96.85 \pm 0.05$ | $90.25 \pm 0.15$ | $73.31 \pm 1.66$ | $95.46 \pm 0.08$ | $85.34 \pm 0.90$ | $60.19 \pm 3.93$ | $91.03 \pm 0.37$ | $83.23 \pm 1.07$ | $59.40 \pm 1.18$ |
| *HNets (1st order)* | $SO(2)$ | $98.87 \pm 0.02$ | $96.20 \pm 0.20$ | $85.03 \pm 1.07$ | $\mathbf{98.75 \pm 0.05}$ | $\mathbf{95.70 \pm 0.12}$ | $\mathbf{82.81 \pm 1.22}$ | $\mathbf{97.61 \pm 0.12}$ | $\mathbf{94.88 \pm 0.06}$ | $79.99 \pm 1.84$ |
| *HNets (2nd order)* | $SO(2)$ | $98.93 \pm 0.08$ | $96.30 \pm 0.06$ | $84.25 \pm 1.26$ | $98.74 \pm 0.01$ | $94.60 \pm 0.18$ | $76.64 \pm 1.45$ | $97.55 \pm 0.08$ | $93.34 \pm 0.65$ | $76.71 \pm 1.98$ |
| *B-CNNs ($k_{max}$)* | $SO(2)$ | $98.90 \pm 0.03$ | $96.20 \pm 0.23$ | $86.12 \pm 1.23$ | $98.64 \pm 0.05$ | $95.57 \pm 0.34$ | $82.58 \pm 0.33$ | $97.14 \pm 0.16$ | $94.66 \pm 0.22$ | $79.54 \pm 2.04$ |
| | $O(2)$ | $97.82 \pm 0.05$ | $92.97 \pm 0.30$ | $77.36 \pm 2.06$ | $97.19 \pm 0.05$ | $92.30 \pm 0.52$ | $73.15 \pm 2.23$ | $93.46 \pm 0.49$ | $89.63 \pm 0.39$ | $68.06 \pm 0.45$ |
| | $O(2)+$ | $99.00 \pm 0.01$ | $96.53 \pm 0.21$ | $87.18 \pm 0.81$ | $\mathbf{98.88 \pm 0.04}$ | $\mathbf{96.41 \pm 0.22}$ | $\mathbf{84.95 \pm 2.29}$ | $97.60 \pm 0.27$ | $\mathbf{95.45 \pm 0.23}$ | $\mathbf{84.85 \pm 1.83}$ |
| | $O(2)+$ | $97.91 \pm 0.01$ | $94.14 \pm 0.08$ | $77.69 \pm 1.75$ | $97.63 \pm 0.03$ | $93.35 \pm 0.09$ | $77.09 \pm 0.81$ | $94.62 \pm 0.36$ | $91.79 \pm 0.08$ | $76.71 \pm 2.85$ |
| *B-CNNs ($k_{max}/2$)* | $SO(2)$ | $98.98 \pm 0.01$ | $96.51 \pm 0.21$ | $\mathbf{87.28 \pm 1.25}$ | $\mathbf{98.89 \pm 0.04}$ | $\mathbf{96.44 \pm 0.34}$ | $\mathbf{86.48 \pm 1.01}$ | $\mathbf{98.16 \pm 0.11}$ | $\mathbf{96.06 \pm 0.09}$ | $\mathbf{86.29 \pm 2.92}$ |

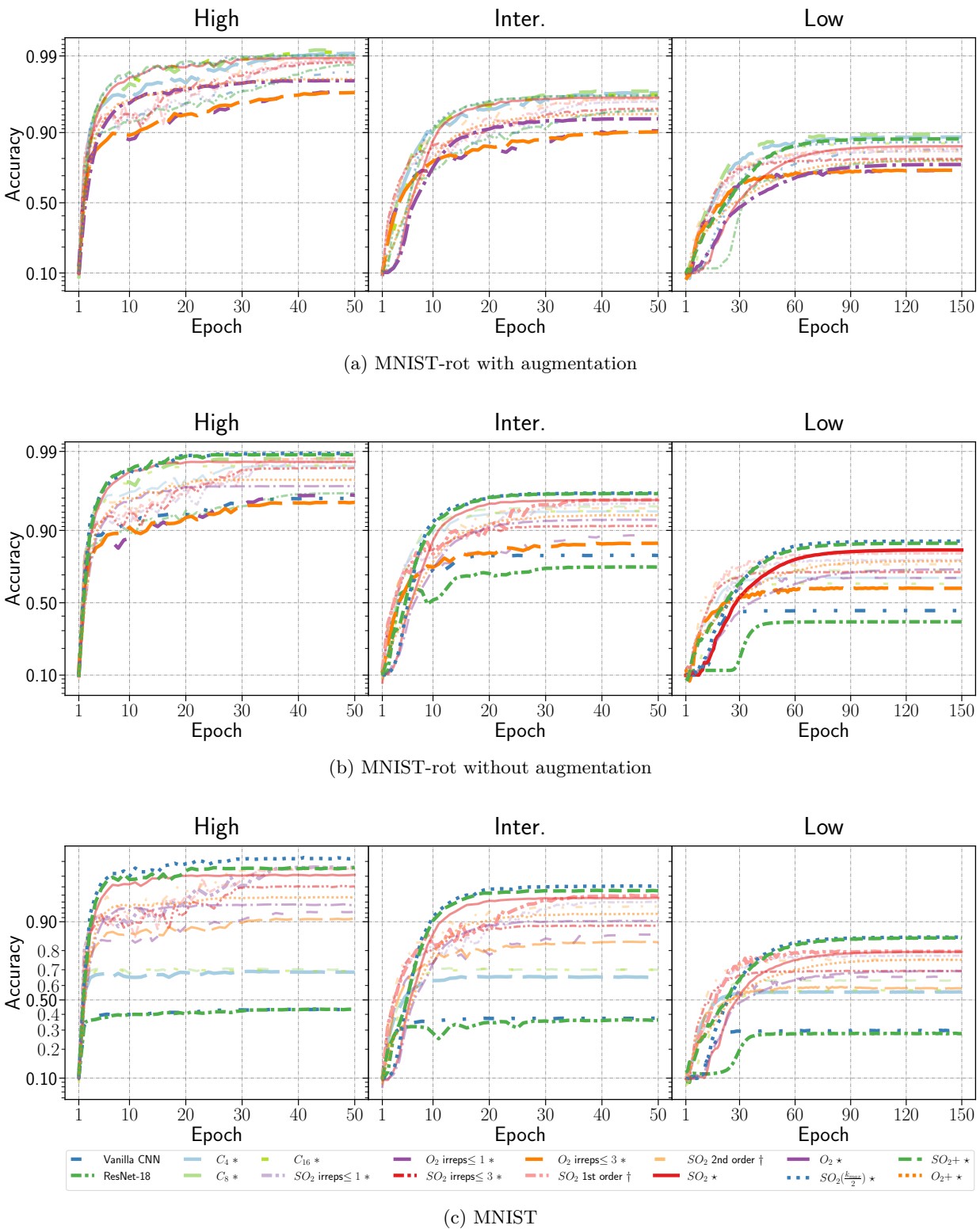

Figure 12: Learning curves obtained for the different methods on the MNIST-rot and MNIST data sets with respect to the epoch. Those learning curves are averaged over 5 independent runs. The legend is the same for all the graphs. The symbols $*$, $\dagger$ and $\star$ refer to the use of $E(2)$-CNNs, HNets and B-CNNs, respectively. Top-3 and worst-3 models are highlighted using full opacity.

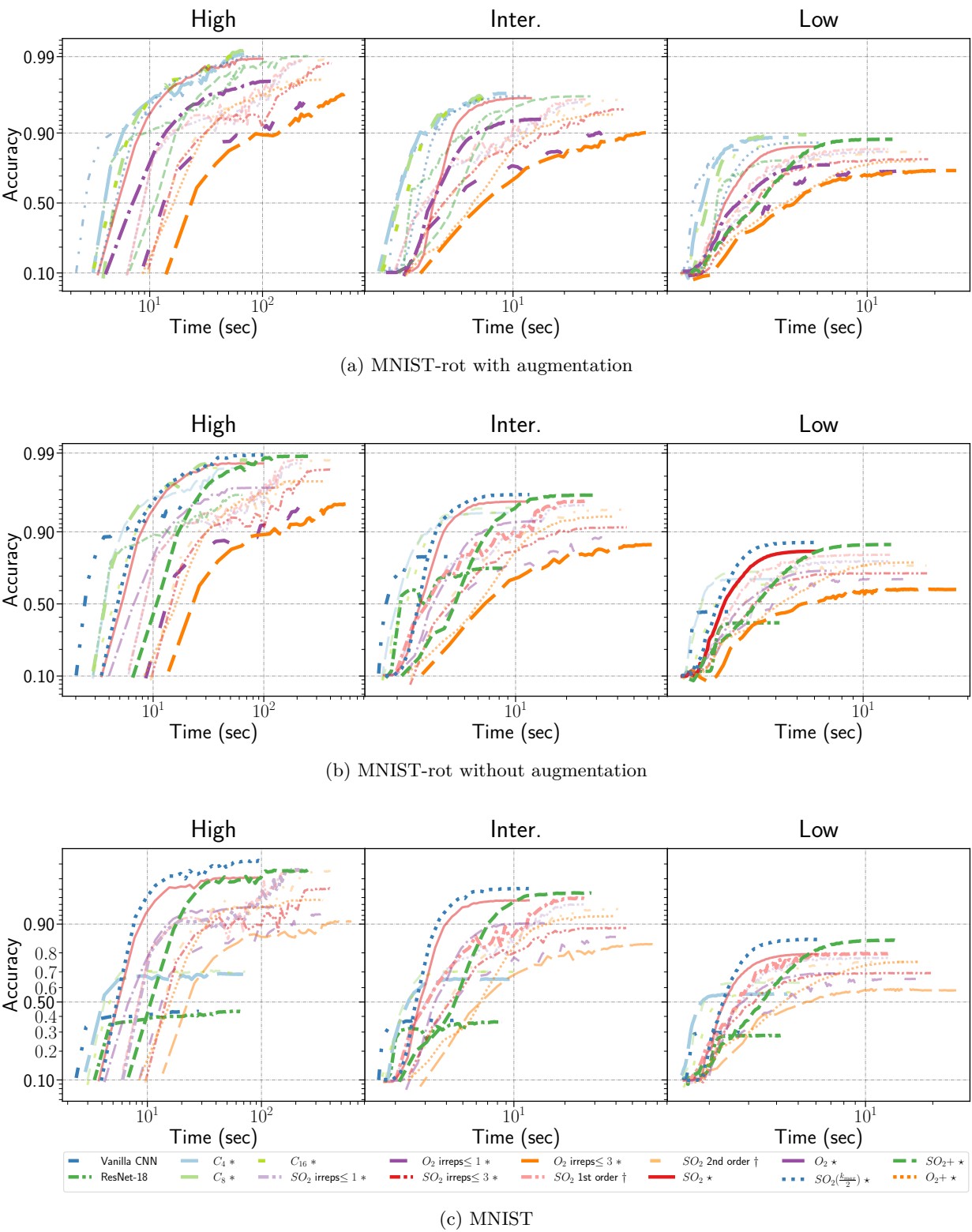

Figure 13: Learning curves obtained for the different methods on the MNIST-rot and MNIST data sets with respect to the wall time (in seconds). Those learning curves are averaged over 5 independent runs. The legend is the same for all the graphs. The symbols $*$, $\dagger$ and $\star$ refer to the use of $E(2)$-CNNs, HNets and B-CNNs, respectively. Top-3 and worst-3 models are highlighted using full opacity.

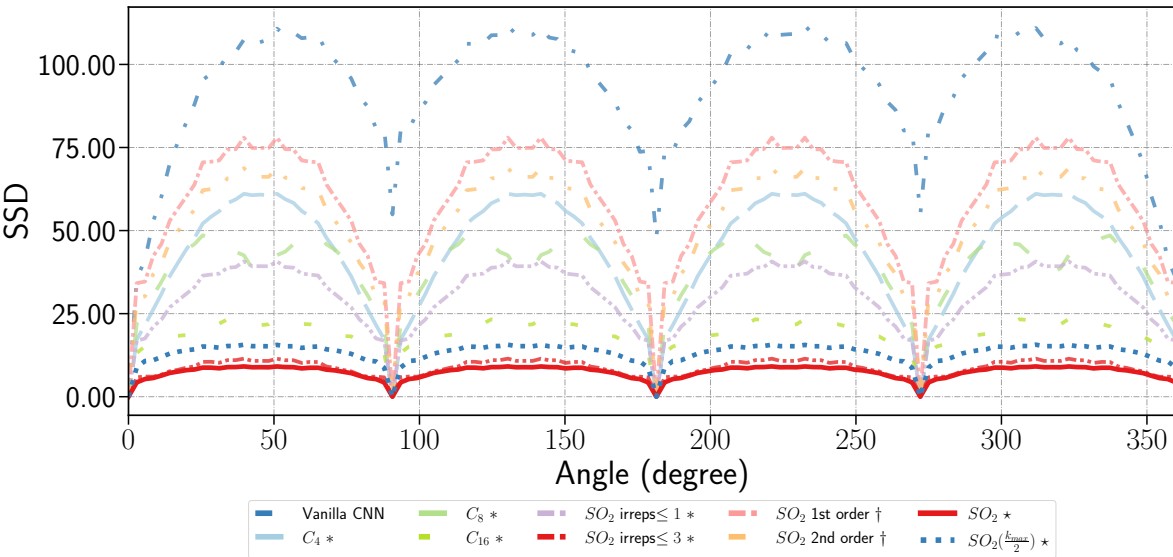

Figure 14: Mean Sum of Squared Differences (SSD) for each image of the MNIST-rot data set. The SSD is computed between the feature maps obtained for the initial images, and all the rotated versions of the same image between 0° to 360° (after being properly re-aligned to allow for pixel-wise comparisons). The symbols $*$, $\dagger$ and $\star$ refer to the use of $E(2)$-CNNs, HNets and B-CNNs, respectively.

Finally, one can observe in Figure 13 that B-CNNs are also a good choice in terms of convergence speed and computational resources, as they are often able to achieve the best performances with a computation time that can be smaller than other methods.

**Equivariance of the feature maps**   For the MNIST-rot data set, a complementary and more qualitative experiment has been conducted in parallel to the previous one. The aim of this experiment is to investigate the global rotation equivariance of the convolutional operation provided by the different methods. To do so, a simple convolutional layer with one feature map for different methods among vanilla CNNs, $E(2)$-CNNs, HNets and B-CNNs have been tested on the whole MNIST-rot data set (without considering any training, only to assess the mathematical equivariance of the operations). For each image, 128 rotated versions between 0° and 360° are considered and forward propagated in the different layers. For each layer, the obtained feature maps are extracted and realigned to the original orientation. By doing so, the Sum of Squared Differences (SSD) can be computed to assess the numerical distance between the feature maps of the original image, and the ones obtained from the rotated versions of the same image.

The results of this additional experiment are presented in Figure 14. On this Figure, one can observe that the rotation equivariance of B-CNNs (and more particularly the $SO(2)$ version with $k_{\max}$) is the best one. It is also really close to the one obtained with the use of $E(2)$-CNNs. For all the different techniques, the equivariance is always significantly better than the one for vanilla CNNs. Nonetheless, one should also mention that interpolations due to the rotations of the feature maps also have an impact on the final SSD.

## 7.4   Results on MNIST(-rot)-back

Table 4 presents the results obtained on the MNIST(-rot)-back data sets, which are variants of the MNIST data set with randomly rotated digits and black and white images as background. For the sake of completeness, Figure 15 and 16 also present all the corresponding training curves with respect to the epoch and the wall time, respectively.

For vanilla CNNs, the conclusions are the same as for MNIST(-rot). Performances quickly drop when using a smaller amount of data, or when data augmentation is not performed properly.

Table 4: Classification accuracy obtained for the different methods on the MNIST-rot-back and MNIST-back data sets, with and without using data augmentation. The *High*, *Inter.* and *Low* data regimes correspond to 12, 000, 1, 200 and 120 images for the training set (same percentage of samples of each target class), respectively. For each column, bold is used to highlight the top-3 performing models. Accuracy and standard deviation are assessed using 5 independent runs. The corresponding training curves are presented in Figure 15 and 16.

| | | MNIST-rot-back | | | | | | MNIST-back | | |
| | | **With data aug.** | | | **Without data aug.** | | | **No rotation during training** | | |
| **Method** | **Group** | High | Inter. | Low | High | Inter. | Low | High | Inter. | Low |
|---|---|---|---|---|---|---|---|---|---|---|
| Vanilla CNN | $\{e\}$ | $89.78 \pm 0.19$ | $75.39 \pm 0.48$ | $41.11 \pm 1.36$ | $78.74 \pm 0.10$ | $38.62 \pm 0.73$ | $20.65 \pm 1.68$ | $35.67 \pm 0.13$ | $26.92 \pm 0.90$ | $18.63 \pm 0.73$ |
| ResNet-18 | $\{e\}$ | $90.05 \pm 0.03$ | $72.64 \pm 0.94$ | $33.39 \pm 1.56$ | $78.86 \pm 0.12$ | $41.02 \pm 0.35$ | $16.62 \pm 0.89$ | $36.16 \pm 0.26$ | $29.35 \pm 0.89$ | $17.13 \pm 0.81$ |
| E(2)-CNN (regular) | $C_4$ | $89.74 \pm 0.25$ | $76.18 \pm 0.69$ | $34.95 \pm 1.56$ | $84.42 \pm 0.21$ | $56.25 \pm 1.90$ | $23.01 \pm 0.14$ | $45.11 \pm 1.12$ | $39.60 \pm 3.02$ | $21.78 \pm 0.65$ |
| | $C_8$ | $\mathbf{90.30 \pm 0.11}$ | $\mathbf{78.46 \pm 0.92}$ | $36.32 \pm 1.41$ | $\mathbf{86.92 \pm 0.32}$ | $58.73 \pm 0.27$ | $22.10 \pm 0.93$ | $45.76 \pm 1.47$ | $43.53 \pm 1.14$ | $23.91 \pm 2.31$ |
| | $C_{16}$ | $89.81 \pm 0.09$ | $76.67 \pm 0.09$ | $36.20 \pm 3.02$ | $85.32 \pm 0.31$ | $47.86 \pm 5.64$ | $19.61 \pm 1.16$ | $44.21 \pm 1.22$ | $40.73 \pm 4.05$ | $21.43 \pm 2.22$ |
| E(2)-CNN (irr. $\leq 1$) | $SO(2)$ | $84.63 \pm 0.33$ | $63.09 \pm 0.67$ | $27.45 \pm 1.68$ | $79.58 \pm 0.47$ | $56.90 \pm 0.64$ | $26.17 \pm 1.33$ | $78.99 \pm 0.26$ | $59.39 \pm 0.89$ | $27.09 \pm 1.41$ |
| | $O(2)$ | $77.17 \pm 1.06$ | $52.64 \pm 0.74$ | $23.43 \pm 0.84$ | $69.47 \pm 1.53$ | $45.54 \pm 1.20$ | $21.35 \pm 1.41$ | $69.25 \pm 0.17$ | $48.37 \pm 0.95$ | $23.85 \pm 0.92$ |
| E(2)-CNN (irr. $\leq 3$) | $SO(2)$ | $84.47 \pm 0.81$ | $65.47 \pm 0.68$ | $34.09 \pm 1.05$ | $79.35 \pm 0.95$ | $53.89 \pm 2.11$ | $26.25 \pm 1.76$ | $77.24 \pm 0.83$ | $53.78 \pm 0.71$ | $28.08 \pm 1.27$ |
| | $O(2)$ | $79.39 \pm 1.11$ | $56.14 \pm 0.42$ | $26.96 \pm 2.15$ | $71.36 \pm 1.38$ | $42.24 \pm 0.72$ | $19.34 \pm 1.84$ | $69.28 \pm 0.65$ | $44.28 \pm 0.30$ | $21.34 \pm 1.17$ |
| HNets (1st order) | $SO(2)$ | $84.80 \pm 0.43$ | $64.24 \pm 1.44$ | $26.25 \pm 0.64$ | $81.92 \pm 0.10$ | $59.92 \pm 1.00$ | $23.76 \pm 1.36$ | $80.35 \pm 0.98$ | $60.33 \pm 2.13$ | $25.70 \pm 1.57$ |
| HNets (2nd order) | $SO(2)$ | $85.72 \pm 0.02$ | $65.61 \pm 0.75$ | $29.34 \pm 3.98$ | $80.93 \pm 0.75$ | $55.38 \pm 2.21$ | $25.22 \pm 0.56$ | $80.49 \pm 0.24$ | $56.75 \pm 0.48$ | $25.84 \pm 2.87$ |
| B-CNNs ($k_{\max}$) | $O(2)$ | $\mathbf{90.99 \pm 0.09}$ | $\mathbf{80.69 \pm 0.35}$ | $39.43 \pm 2.00$ | $\mathbf{89.00 \pm 0.24}$ | $\mathbf{78.83 \pm 0.62}$ | $\mathbf{33.98 \pm 6.53}$ | $\mathbf{88.43 \pm 0.54}$ | $\mathbf{81.12 \pm 0.88}$ | $\mathbf{35.38 \pm 4.03}$ |
| | $SO(2)$ | $88.25 \pm 0.21$ | $76.51 \pm 0.38$ | $\mathbf{40.02 \pm 1.30}$ | $86.18 \pm 0.10$ | $73.86 \pm 0.32$ | $32.65 \pm 7.76$ | $83.16 \pm 0.15$ | $73.23 \pm 1.43$ | $38.62 \pm 3.87$ |
| | $SO(2)+$ | $\mathbf{90.37 \pm 0.19}$ | $\mathbf{81.24 \pm 0.07}$ | $30.33 \pm 8.13$ | $\mathbf{89.35 \pm 0.09}$ | $\mathbf{80.08 \pm 0.48}$ | $28.63 \pm 6.19$ | $\mathbf{88.17 \pm 0.65}$ | $\mathbf{80.16 \pm 0.27}$ | $\mathbf{35.10 \pm 6.25}$ |
| B-CNNs ($k_{\max}/2$) | $O(2)+$ | $87.88 \pm 0.19$ | $75.99 \pm 1.06$ | $30.92 \pm 1.30$ | $86.59 \pm 0.04$ | $74.50 \pm 0.29$ | $\mathbf{32.84 \pm 1.18}$ | $84.48 \pm 0.25$ | $75.08 \pm 1.19$ | $32.71 \pm 3.81$ |
| | $SO(2)$ | $87.53 \pm 0.16$ | $77.35 \pm 0.46$ | $\mathbf{37.04 \pm 2.76}$ | $86.50 \pm 0.15$ | $\mathbf{76.21 \pm 0.29}$ | $\mathbf{33.65 \pm 1.83}$ | $\mathbf{85.73 \pm 0.18}$ | $\mathbf{78.95 \pm 0.29}$ | $31.06 \pm 3.71$ |

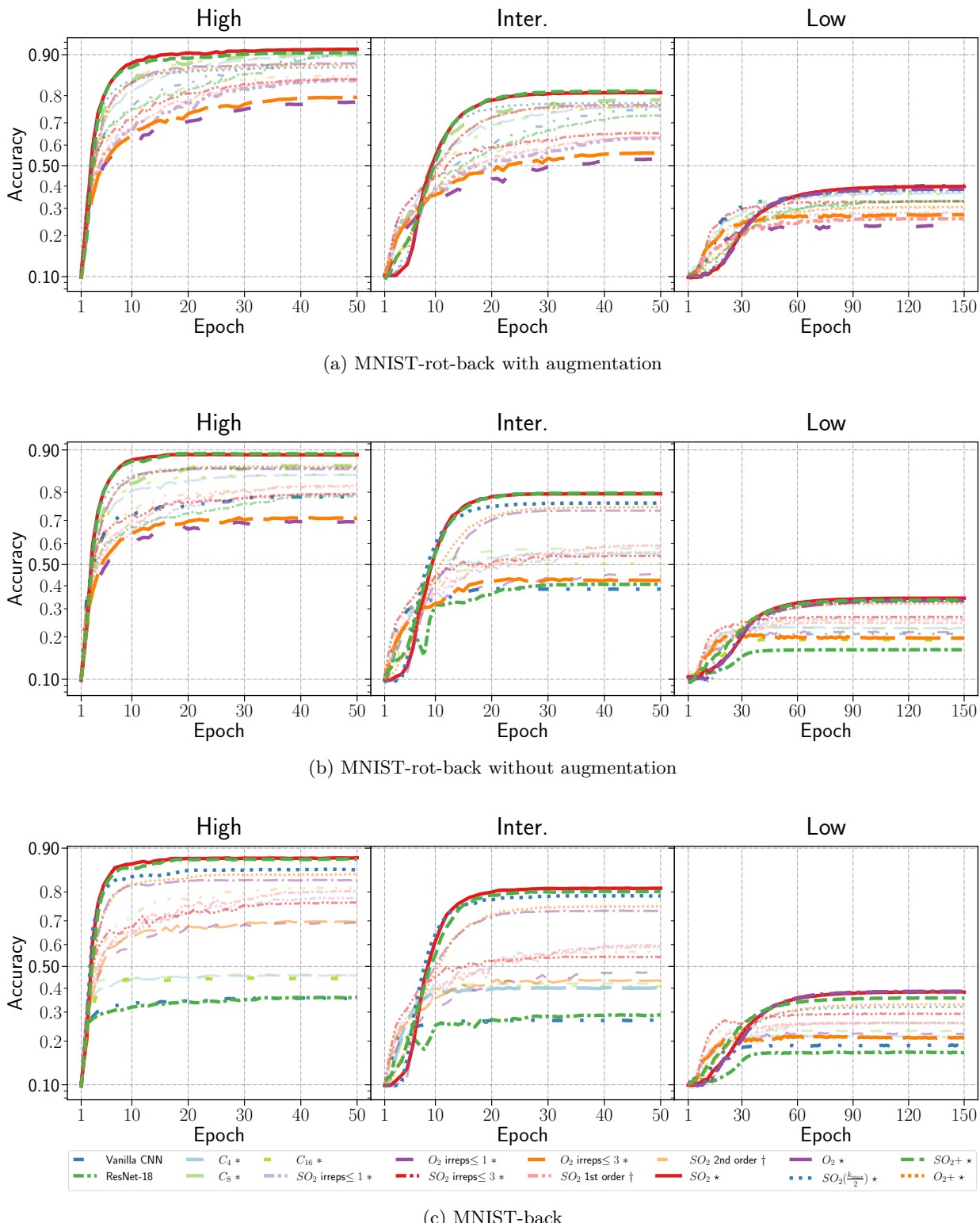

Figure 15: Learning curves obtained for the different methods on the MNIST-back and MNIST-rot-back data sets with respect to the epoch. Those learning curves are averaged over 5 independent runs. The legend is the same for all the graphs. The symbols $*$, $\dagger$ and $\star$ refer to the use of $E(2)$-CNNs, HNets and B-CNNs, respectively. Top-3 and worst-3 models are highlighted using full opacity.

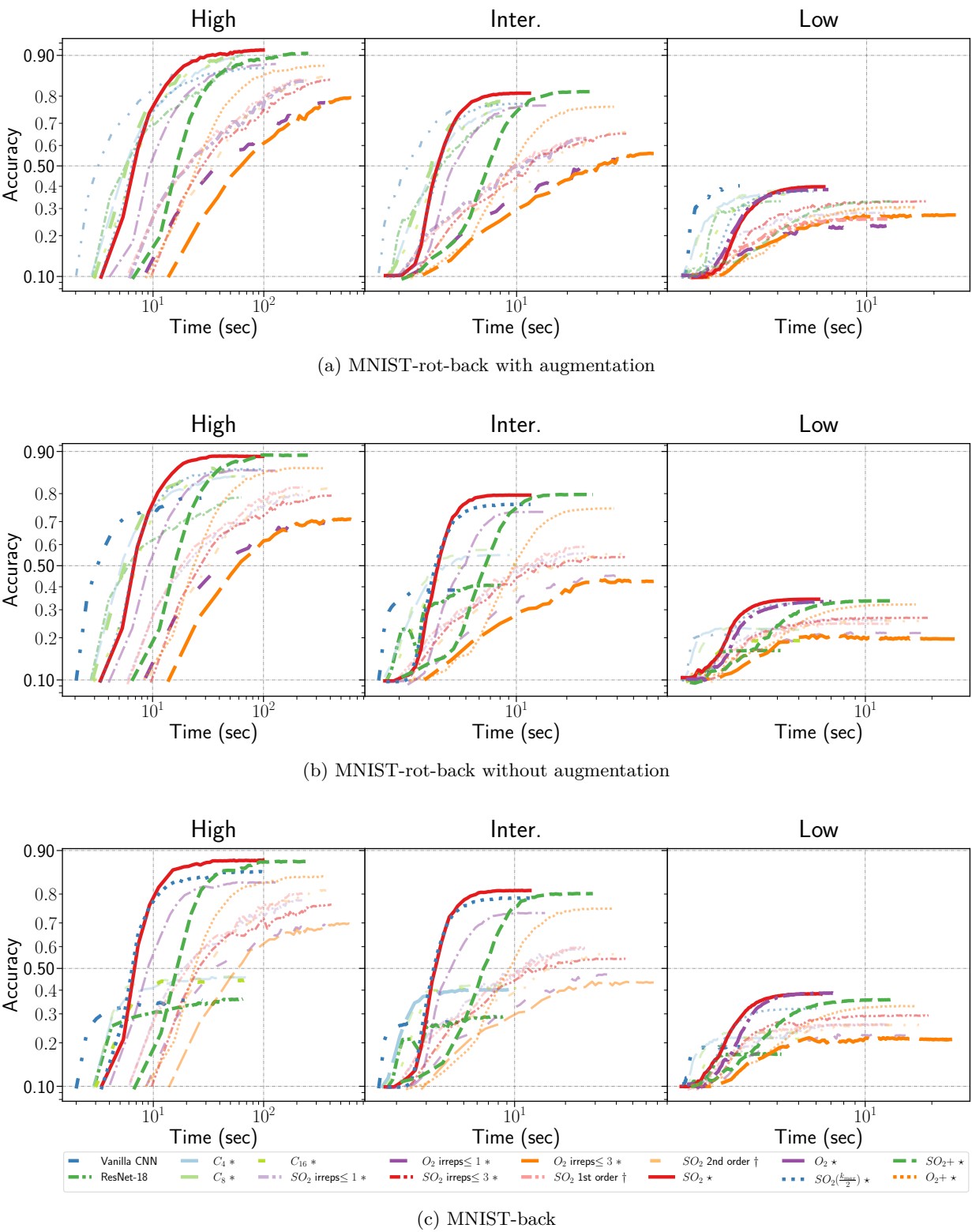

(a) MNIST-rot-back with augmentation

(b) MNIST-rot-back without augmentation

(c) MNIST-back

Figure 16: Learning curves obtained for the different methods on the MNIST-back and MNIST-rot-back data sets with respect to the wall time (in seconds). Those learning curves are averaged over 5 independent runs. The legend is the same for all the graphs. The symbols $*$, $\dagger$ and $\star$ refer to the use of $E(2)$-CNNs, HNets and B-CNNs, respectively. Top-3 and worst-3 models are highlighted using full opacity.

Now, by opposition to the observation for the MNIST(-rot) data set, it appears that B-CNNs are from a general point of view significantly better than other equivariant models, in each setup. For the high data setting on MNIST-back (without seeing rotated images during training), Figure 15 clearly reveals several groups of plateau corresponding to vanilla CNNs, discrete $E(2)$-CNNs, $O(2)$ $E(2)$-CNNs, $SO(2)$ $E(2)$-CNNs and HNets, and finally all the B-CNNs models with the $SO(2)$ ones being in top of them. Again, in addition to the better performances, one should also highlight a faster convergence for the B-CNNs. B-CNNs are able to achieve better performances, in a computation time that is equivalent or even slightly better.

## 7.5  Results on Galaxy10 DECals

Table 5 presents the results obtained on the Galaxy10 DECals data set. For the sake of completeness, Figure 17 and 18 also present all the corresponding training curves with respect to the epoch and the wall time, respectively.

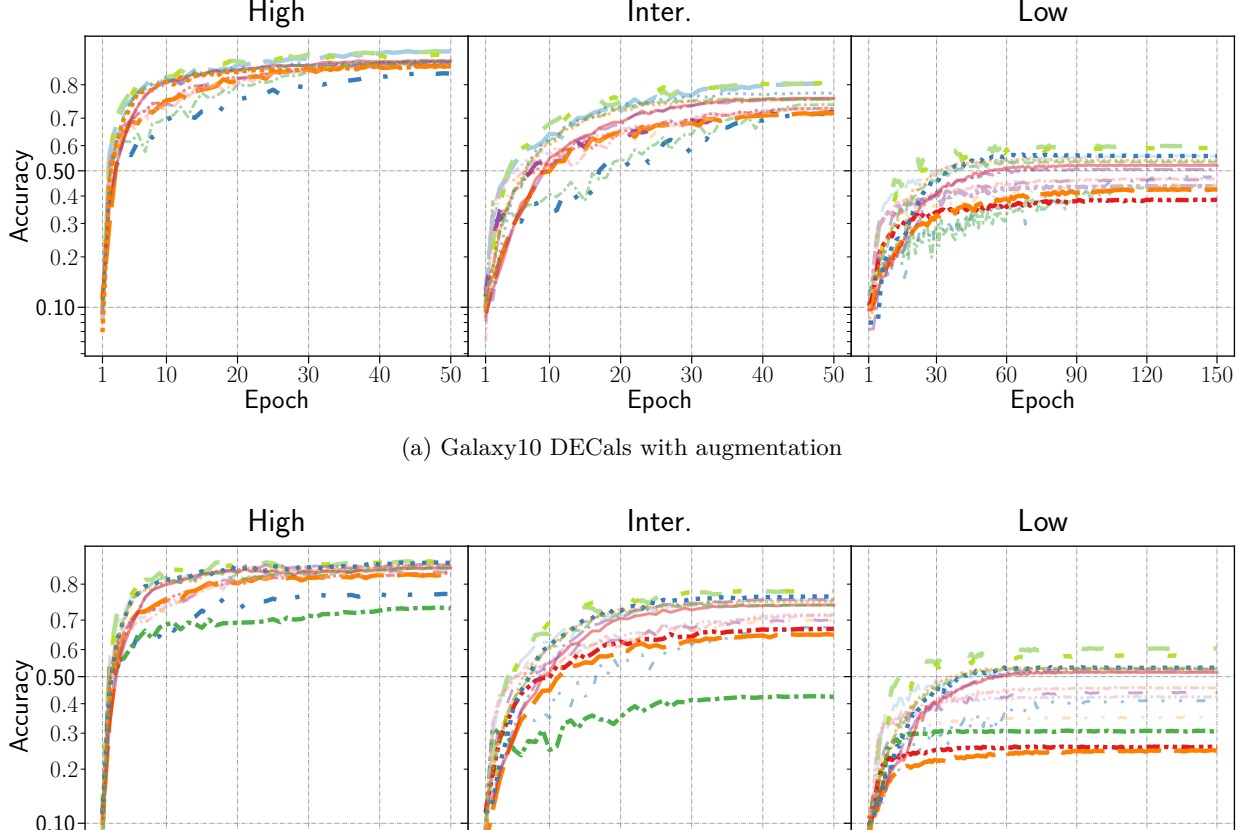

(a) Galaxy10 DECals with augmentation

(b) Galaxy10 DECals without augmentation

Figure 17: Learning curves obtained for the different methods on the Galaxy10 DECals data set with respect to the epoch. Those learning curves are averaged over 5 independent runs. The legend is the same for all the graphs. The symbols $*$, $\dagger$ and $\star$ refer to the use of $E(2)$-CNNs, HNets and B-CNNs, respectively. Top-3 and worst-3 models are highlighted using full opacity.

Table 5: Classification accuracy obtained for the different methods on the Galaxy10 DECals data set, with and without using data augmentation. The *High*, *Inter.* and *Low* data regimes correspond to 14,188, 1,418 and 141 images for the training set (same percentage of samples of each target class), respectively. For each column, bold is used to highlight the top-3 performing models. Accuracy and standard deviation are assessed using 5 independent runs. The corresponding training curves are presented in Figure 17 and 18.

| Method | Group | Galaxy10 DECals | | | | | |
| | | With data aug. | | | Without data aug. | | |
| | | High | Inter. | Low | High | Inter. | Low |
|---|---|---|---|---|---|---|---|
| Vanilla CNN | {e} | $82.98 \pm 0.57$ | $71.67 \pm 0.62$ | $46.75 \pm 2.87$ | $77.33 \pm 0.44$ | $68.32 \pm 0.08$ | $41.47 \pm 1.51$ |
| ResNet-18 | {e} | $85.70 \pm 0.41$ | $74.45 \pm 0.39$ | $44.22 \pm 1.26$ | $73.76 \pm 0.10$ | $42.15 \pm 0.85$ | $31.37 \pm 0.47$ |
| E(2)-CNN (regular) | $C_4$ | $\mathbf{87.32 \pm 0.40}$ | $\mathbf{80.32 \pm 0.39}$ | $\mathbf{55.70 \pm 1.89}$ | $\mathbf{83.86 \pm 0.24}$ | $\mathbf{76.28 \pm 0.26}$ | $52.68 \pm 0.78$ |
| | $C_8$ | $\mathbf{87.46 \pm 0.63}$ | $\mathbf{80.51 \pm 0.49}$ | $\mathbf{60.50 \pm 3.46}$ | $\mathbf{84.54 \pm 0.23}$ | $\mathbf{78.02 \pm 0.09}$ | $\mathbf{59.18 \pm 1.05}$ |
| | $C_{16}$ | $\mathbf{86.66 \pm 0.32}$ | $\mathbf{80.64 \pm 0.39}$ | $\mathbf{58.82 \pm 2.72}$ | $\mathbf{84.70 \pm 0.49}$ | $\mathbf{78.57 \pm 0.59}$ | $\mathbf{57.23 \pm 6.58}$ |
| E(2)-CNN (irr. ≤ 1) | $SO(2)$ | $85.12 \pm 0.51$ | $73.32 \pm 0.34$ | $42.48 \pm 0.67$ | $83.81 \pm 0.06$ | $71.42 \pm 0.37$ | $41.89 \pm 1.53$ |
| | $O(2)$ | $84.78 \pm 0.23$ | $72.13 \pm 0.17$ | $48.61 \pm 2.54$ | $82.97 \pm 0.67$ | $69.53 \pm 0.33$ | $43.41 \pm 1.27$ |
| E(2)-CNN (irr. ≤ 3) | $SO(2)$ | $85.14 \pm 0.19$ | $72.82 \pm 0.66$ | $39.37 \pm 2.64$ | $82.21 \pm 0.54$ | $66.56 \pm 0.94$ | $26.62 \pm 1.04$ |
| | $O(2)$ | $83.99 \pm 0.87$ | $71.54 \pm 1.10$ | $43.73 \pm 4.08$ | $82.30 \pm 0.22$ | $65.58 \pm 1.54$ | $25.14 \pm 0.80$ |
| HNets (1st order) | $SO(2)$ | $85.37 \pm 0.48$ | $73.20 \pm 0.34$ | $46.58 \pm 2.49$ | $83.91 \pm 0.32$ | $71.49 \pm 0.31$ | $45.77 \pm 1.82$ |
| HNets (2nd order) | $SO(2)$ | $84.91 \pm 0.67$ | $72.77 \pm 0.16$ | $44.53 \pm 0.12$ | $82.58 \pm 0.44$ | $69.49 \pm 0.79$ | $34.81 \pm 3.17$ |
| B-CNNs ($k_{max}$) | $SO(2)$ | $85.46 \pm 0.54$ | $76.02 \pm 0.59$ | $52.89 \pm 2.56$ | $83.46 \pm 0.12$ | $74.46 \pm 0.71$ | $52.29 \pm 3.33$ |
| | $O(2)$ | $85.14 \pm 0.38$ | $76.41 \pm 0.17$ | $49.75 \pm 0.63$ | $84.13 \pm 0.21$ | $76.02 \pm 0.62$ | $54.39 \pm 3.87$ |
| | $SO(2)+$ | $85.20 \pm 0.38$ | $75.63 \pm 0.76$ | $51.73 \pm 1.60$ | $83.53 \pm 0.18$ | $74.38 \pm 0.11$ | $52.28 \pm 2.21$ |
| B-CNNs ($k_{max}/2$) | $O(2)+$ | $84.39 \pm 0.77$ | $75.62 \pm 0.43$ | $52.81 \pm 0.85$ | $84.42 \pm 0.15$ | $75.37 \pm 0.81$ | $53.32 \pm 2.34$ |
| | $SO(2)$ | $85.65 \pm 0.27$ | $77.73 \pm 0.50$ | $54.31 \pm 2.47$ | $\mathbf{84.94 \pm 0.32}$ | $76.53 \pm 1.04$ | $\mathbf{54.76 \pm 3.16}$ |

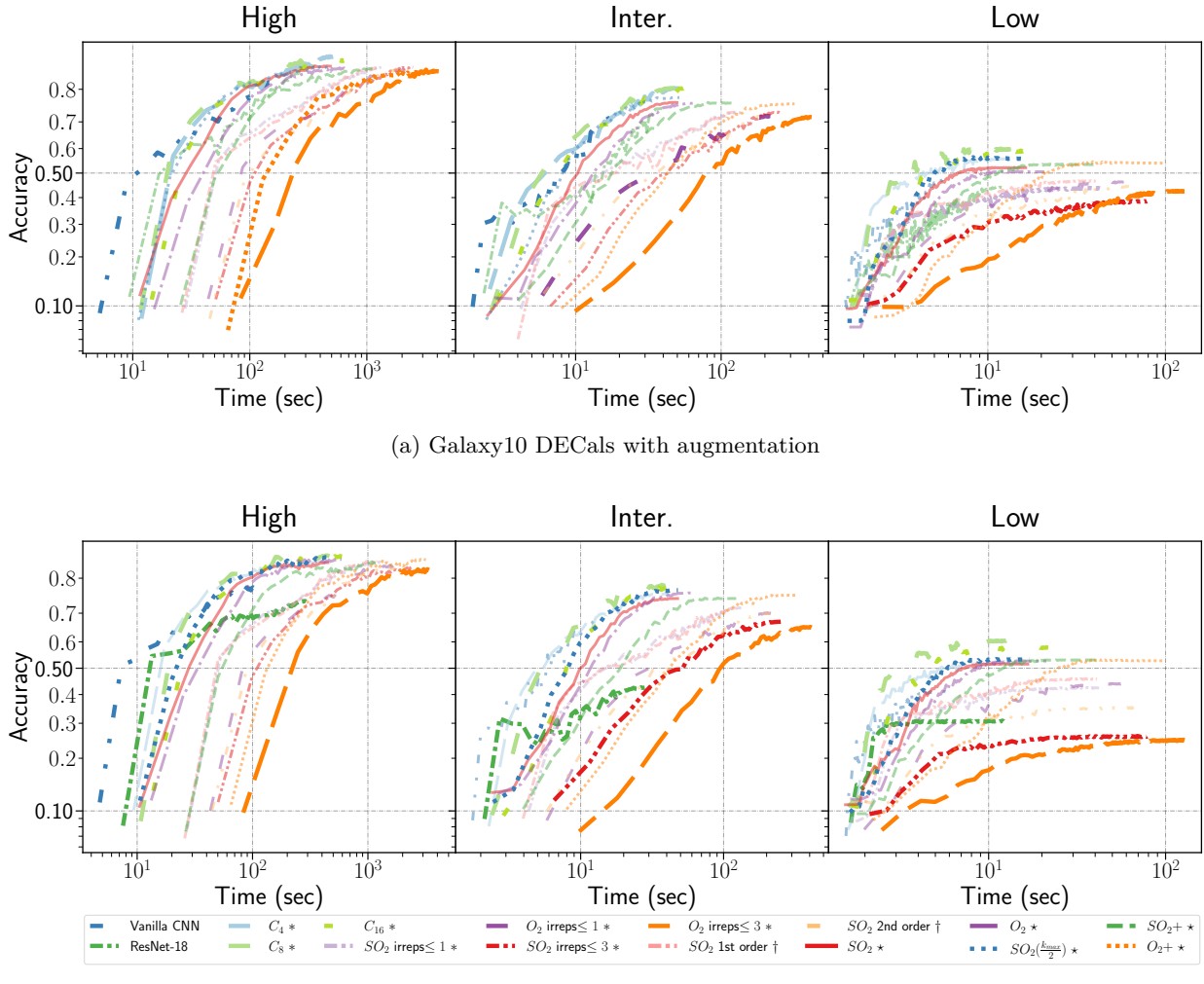

Figure 18: Learning curves obtained for the different methods on the Galaxy10 DECals data set with respect to the wall time (in seconds). Those learning curves are averaged over 5 independent runs. The legend is the same for all the graphs. The symbols $*$, $\dagger$ and $\star$ refer to the use of $E(2)$-CNNs, HNets and B-CNNs, respectively. Top-3 and worst-3 models are highlighted using full opacity.

From those results, one can see again that using the symmetry group $C_4$, $C_8$ or $C_{16}$ already allow $E(2)$-CNNs to achieve very good results. For Galaxy10 DECals, this observation stands for each setup, even without using data augmentation.

$SO(2)$-based B-CNNs with the low cutoff policy ($k_{\max}/2$) is again one of the best performing model when used without data augmentation.

It is interesting to see that using the $O(2)$ group does not always lead to better performances compared to results obtained using $SO(2)$, despite the fact planar reflections are meaningful for this application.

### 7.6 Results on Malaria

Table 6 presents the results obtained on the Malaria data set. For the sake of completeness, Figure 19 and 20 also present all the corresponding training curves with respect to the epoch and the wall time, respectively.

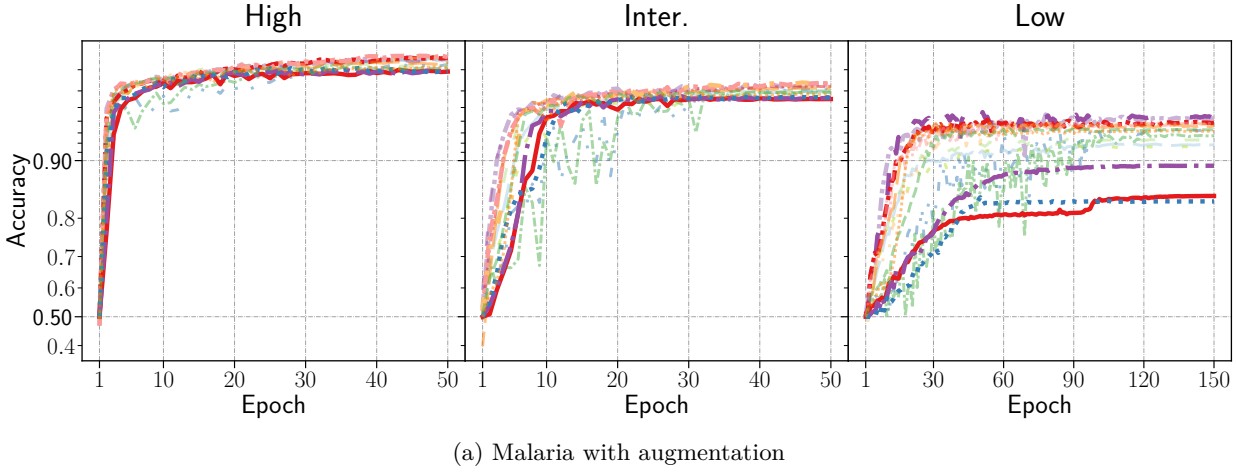

(a) Malaria with augmentation

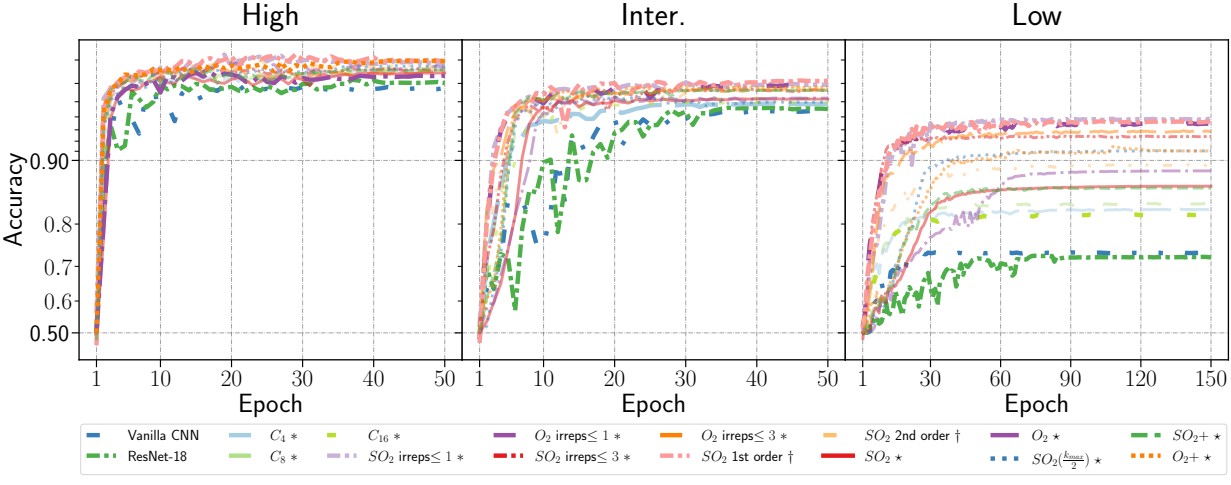

(b) Malaria without augmentation

Figure 19: Learning curves obtained for the different methods on the Malaria data set with respect to the epoch. Those learning curves are averaged over 5 independent runs. The legend is the same for all the graphs. The symbols $*$, $\dagger$ and $\star$ refer to the use of $E(2)$-CNNs, HNets and B-CNNs, respectively. Top-3 and worst-3 models are highlighted using full opacity.

Interestingly, for this data set, B-CNNs seem to perform slightly worse than other methods. With the use of data augmentation, $E(2)$-CNNs and HNets are the best performing models. B-CNNs are only able to be

Table 6: Classification accuracy obtained for the different methods on the Malaria data set, with and without using data augmentation. The *High*, *Inter.* and *Low* data regimes correspond to 22,046, 2,204 and 220 images for the training set (same percentage of samples of each target class), respectively. For each column, bold is used to highlight the top-3 performing models. Accuracy and standard deviation are assessed using 5 independent runs. The corresponding training curves are presented in Figure 19 and 20.

| | | Malaria | | | | | |
| | | With data aug. | | | Without data aug. | | |
| **Method** | **Group** | High | Inter. | Low | High | Inter. | Low |
|---|---|---|---|---|---|---|---|
| Vanilla CNN | $\{e\}$ | $97.07 \pm 0.05$ | $96.09 \pm 0.06$ | $93.36 \pm 0.34$ | $95.69 \pm 0.15$ | $94.47 \pm 0.09$ | $73.36 \pm 2.63$ |
| ResNet-18 | $\{e\}$ | $97.41 \pm 0.13$ | $96.05 \pm 0.11$ | $93.29 \pm 0.24$ | $95.99 \pm 0.10$ | $94.55 \pm 0.23$ | $68.81 \pm 3.13$ |
| E(2)-CNN (*regular*) | $C_4$ | $97.31 \pm 0.19$ | $96.05 \pm 0.24$ | $91.96 \pm 0.74$ | $96.53 \pm 0.11$ | $94.81 \pm 0.22$ | $81.57 \pm 2.16$ |
| | $C_8$ | $97.22 \pm 0.15$ | $95.87 \pm 0.16$ | $92.43 \pm 0.43$ | $96.59 \pm 0.20$ | $95.08 \pm 0.21$ | $82.53 \pm 1.86$ |
| | $C_{16}$ | $96.86 \pm 0.03$ | $95.64 \pm 0.24$ | $91.19 \pm 0.46$ | $96.63 \pm 0.08$ | $94.92 \pm 0.32$ | $82.24 \pm 2.59$ |
| E(2)-CNN (*irr.* $\leq 1$) | $SO(2)$ | $\mathbf{97.44 \pm 0.12}$ | $\mathbf{96.32 \pm 0.13}$ | $\mathbf{94.44 \pm 0.38}$ | $\mathbf{96.76 \pm 0.15}$ | $\mathbf{95.89 \pm 0.12}$ | $\mathbf{94.21 \pm 0.25}$ |
| | $O(2)$ | $97.25 \pm 0.26$ | $96.12 \pm 0.05$ | $\mathbf{94.13 \pm 0.37}$ | $96.68 \pm 0.07$ | $\mathbf{95.85 \pm 0.20}$ | $\mathbf{94.12 \pm 0.70}$ |
| E(2)-CNN (*irr.* $\leq 3$) | $SO(2)$ | $\mathbf{97.43 \pm 0.20}$ | $96.28 \pm 0.06$ | $93.95 \pm 0.49$ | $96.56 \pm 0.11$ | $95.53 \pm 0.12$ | $\mathbf{93.08 \pm 0.86}$ |
| | $O(2)$ | $97.27 \pm 0.26$ | $96.19 \pm 0.10$ | $94.00 \pm 0.73$ | $96.47 \pm 0.22$ | $95.56 \pm 0.19$ | $92.49 \pm 1.26$ |
| HNets (*1st order*) | $SO(2)$ | $\mathbf{97.43 \pm 0.16}$ | $\mathbf{96.41 \pm 0.02}$ | $93.89 \pm 0.88$ | $\mathbf{96.85 \pm 0.16}$ | $\mathbf{96.13 \pm 0.14}$ | $93.87 \pm 0.61$ |
| HNets (*2nd order*) | $SO(2)$ | $97.43 \pm 0.20$ | $\mathbf{96.43 \pm 0.01}$ | $\mathbf{94.08 \pm 0.33}$ | $96.76 \pm 0.21$ | $95.77 \pm 0.08$ | $87.92 \pm 4.11$ |
| B-CNNs ($k_{\max}$) | $SO(2)$ | $96.81 \pm 0.18$ | $95.53 \pm 0.23$ | $88.80 \pm 4.20$ | $96.39 \pm 0.16$ | $95.16 \pm 0.25$ | $84.74 \pm 2.95$ |
| | $O(2)$ | $96.86 \pm 0.06$ | $95.54 \pm 0.21$ | $88.70 \pm 1.52$ | $96.33 \pm 0.25$ | $95.16 \pm 0.39$ | $88.07 \pm 3.69$ |
| | $SO(2)+$ | $96.97 \pm 0.21$ | $95.91 \pm 0.09$ | $93.69 \pm 0.45$ | $96.51 \pm 0.08$ | $95.48 \pm 0.07$ | $89.83 \pm 2.16$ |
| B-CNNs ($k_{\max}/2$) | $O(2)+$ | $97.01 \pm 0.04$ | $95.79 \pm 0.24$ | $93.69 \pm 0.75$ | $\mathbf{96.82 \pm 0.23}$ | $\mathbf{95.78 \pm 0.08}$ | $90.81 \pm 0.51$ |
| | $SO(2)$ | $96.83 \pm 0.21$ | $95.58 \pm 0.28$ | $92.19 \pm 0.87$ | $96.71 \pm 0.09$ | $95.06 \pm 0.11$ | $91.65 \pm 1.21$ |

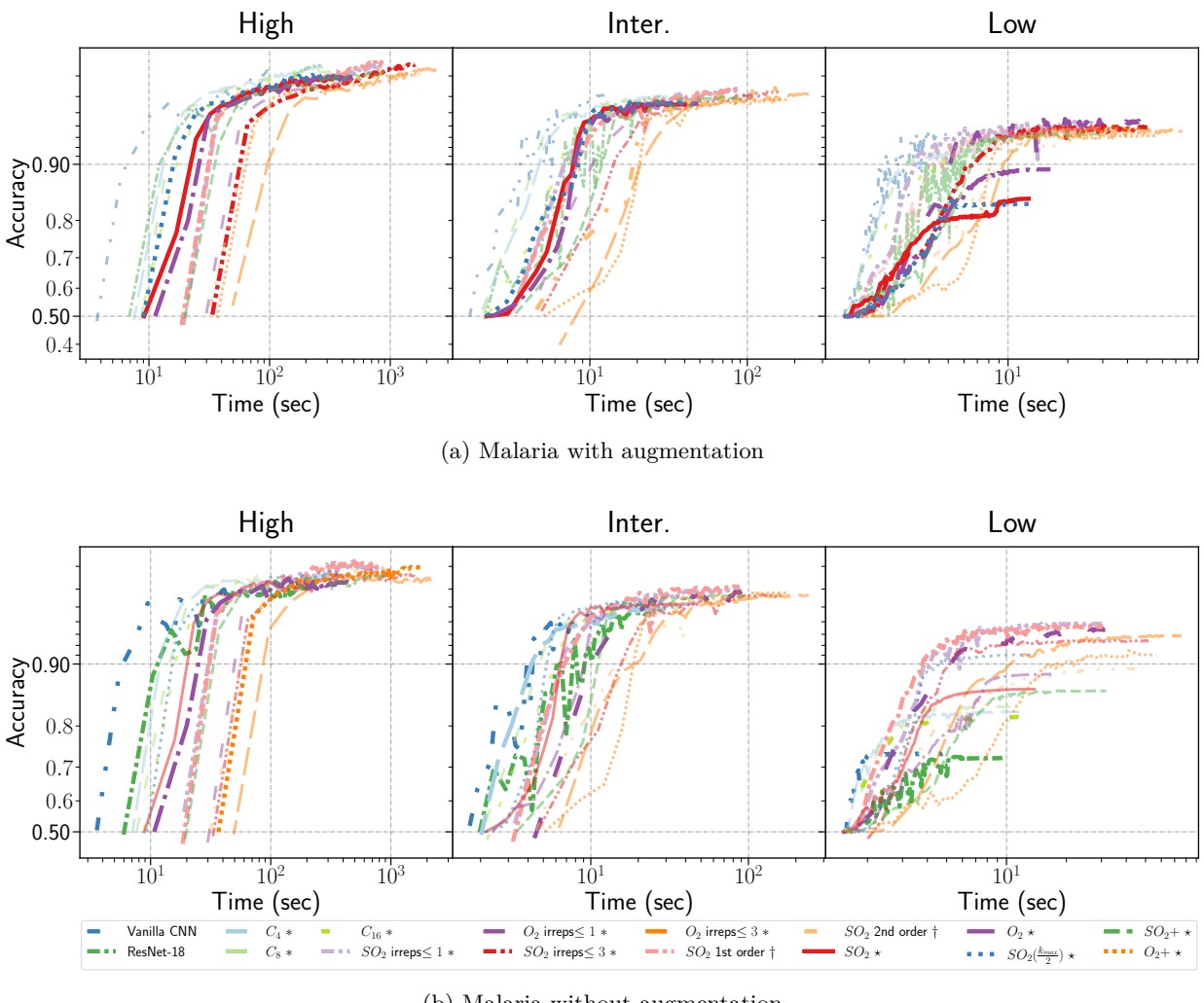

Figure 20: Learning curves obtained for the different methods on the Malaria data set with respect to the wall time (in seconds). Those learning curves are averaged over 5 independent runs. The legend is the same for all the graphs. The symbols ∗, † and ⋆ refer to the use of $E(2)$-CNNs, HNets and B-CNNs, respectively. Top-3 and worst-3 models are highlighted using full opacity.

competitive without the use of data augmentation. This highlights the fact that $(S-)O(2)$ invariance may be less useful than for the other tested applications. Still, performances are most of the time very close to each other.

# 8 Discussion

This section first provides a global discussion regarding the different experiments and results presented in the previous section. Then, we discuss the choice of using models with automatic symmetry discovery, or models as B-CNNs based on applying (strong) constraints to guarantee user-defined symmetry.

## 8.1 Global Discussion of the Results

From the experiments and the preliminary discussions in the previous section, several insights can be retrieved.

**Equivariant models vs. vanilla CNNs**  In the particular case where many data are available, vanilla CNNs do a very decent job by being only marginally below the top-accuracy. Thanks to data augmentation, those models seem to be able to learn meaningful invariances. However, by taking a look at the training curves, it appears clear that convergence is much slower. This is easily explained by the fact that vanilla CNNs should learn the invariances, while it is not the case for equivariant models. Therefore, computation time and energy may be saved by using instead an equivariant model with training through much less epochs. Furthermore, this drawback is emphasized when vanilla CNNs should work with less data and/or without data augmentation, up to leading to very poor performances in those cases.

**Discrete vs. continuous groups**  As already spotted by Weiler & Cesa (2019), using discrete groups already largely improve performances, and even sometimes constitute the best performing models. Nonetheless, experiments also highlight that it does not guarantee equivariance and often still requires a larger amount of data as well as data augmentation.

**B-CNNs vs. other equivariant models**  In our experiments, B-CNNs are most of the time at least able to achieve state-of-the-art performances. In low data settings, they are often the best performing models. In particular, B-CNNs with low cutoff policies ($k_{\max}/2$) seem very efficient. They achieve top-1 accuracy in 7 setups and top-3 accuracy in 14 setups, among a total of 30 different setups (all the columns for all the data sets). Now, by considering all the B-CNNs model at the same time, they achieve together top-1 accuracy in 16 setups and top-3 accuracy in 20 setups. For comparison, it is better than $E(2)$-CNNs and HNets, which achieve top-1 accuracy in 11 (11 for the discrete versions and 0 for the continuous groups) and 3 setups, and top-3 accuracy in 19 (12 for the discrete versions and 7 for the continuous groups) and 10 setups, respectively. Also, from the experiments on MNIST and MNIST-back (no rotation during training), one can see that the invariance achieved by design in B-CNNs is better than for other methods as performances for example for the low cutoff policy are significantly above. Finally, we can observe that B-CNNs are able to achieve good performances even without data augmentation, which is not/less the case for other methods. As data augmentation increases the computational cost of training (because of the increase of the number of training data and/or the number of training epochs), B-CNNs may therefore be a more favorable approach.

From a general point of view, an interesting conclusion of this work is also that using continuous equivariance is not necessarily always the best performing approach in terms of accuracy. However, when it is not the case, using $SO(2)$ or $O(2)$ equivariance could still be useful in terms of computational resources (convergence can be faster, and less epoch are therefore required) or in order to apply constraints that are required by the domain expert. On their side, B-CNNs have the advantage of being simple to use while being at least competitive with the state-of-the-art performances.

### 8.2 Automatic Symmetry Discovery vs. Constraints-based Models

This work focuses on constraint-based models that assume that users know *a priori* the appropriate symmetry group(s) for the application at hand. However, it can sometimes be hard to obtain this prior knowledge as it requires good understanding of the data/application. Methods like the one proposed by Dehmamy et al. (2021) (L-CNNs) therefore attempt to infer the invariance(s) that need to be enforced automatically during training. Here, we advocate that the constraint-based approach remains relevant and often competitive.

Firstly, B-CNNs and other constraint-based methods can be adapted to handle symmetries in a data-driven fashion. For example, one can consider a method similar to the one in Section 6.4 to let the model choose meaningful symmetries. By designing the architecture with multiple networks in parallel that provide different invariances (one could simultaneously consider $SO(2)$, $O(2)$, $SO(2)+$ and $O(2)+$), the model can benefit from multiple views of the problem and use the features that are the most relevant. In a data-driven fashion, the relevant part(s) of the network will be retained so as to enforce appropriate invariance(s). This approach does not rely on the hypothetical ability of vanilla CNNs to learn specific types of invariance, but rather builds on models that are designed for that.

Secondly, B-CNNs and other constraint-based methods have the advantage to guarantee specific invariances. Instead of using data augmentation and relying on a proper learning of the invariances, mathematically sound mechanisms are used, such as Bessel coefficients for B-CNNs. However, these mechanisms can only deal with an invariance that can be described with reasonable mathematical complexity. Yet, handling symmetries in a data-driven fashion (discovering useful symmetries during training, with vanilla CNNs or other more adapted methods like L-CNNs) is not a one-fits-all solution and some invariances may be impossible to learn without additional mechanisms.

Thirdly, the way constraints are enforced in B-CNNs allows them to exhibit an invariance that can even not be present in the training data set. Hence, as shown in the above experiments, B-CNNs do not rely on data augmentation that increases the computational cost[5], nor do they require to see training data with different orientations for the rotation invariance. This is a consequence of the mathematical soundness of the approach used in B-CNNs.

To conclude, automatic symmetry discovery and constraints-based models are two paradigms that should be used in different situations. While automatic symmetry discovery is useful when no prior knowledge is available and the invariance may be complex to describe mathematically, it relies on a appropriate learning of the invariances that may fail. On the other side, constraints-based models like B-CNNs require a prior knowledge of the invariances involved in the applications, but can provide strong guarantees.

## 9 Conclusion and Future Work

This work provides a comprehensive explanation of B-CNNs, including their mathematical foundations and key findings. Improvements are presented and compared to the prior work of Delchevalerie et al. (2021), including making B-CNNs also equivariant to reflections and multi-scale. Furthermore, the previous troublesome meta-parameters $m_{\max}$ and $j_{\max}$ that were hard to fine-tune have been replaced with a single meta-parameter $k_{\max}$ for which an optimal choice can be computed using the Nyquist frequency.

An extensive empirical study has been conducted to assess the performance of B-CNNs compared to already existing techniques. One can conclude that B-CNNs have, most of the time, better performances than the other state-of-the-art methods, and achieve in the worst cases roughly the same performances. In low data settings, they actually outperform other models most of the time. This is mainly due to the B-CNNs ability to maintain robust invariances without resorting to data augmentation techniques, which is often not the case for other models. Furthermore, we show that B-CNNs exhibit a more systematic and mathematically rigorous equivariance, which can be meaningful for many applications. Finally, B-CNNs do not involve particular, more exotic (such as complex-valued feature maps), representations for feature maps and are therefore highly compatible with already existing deep learning techniques and frameworks.

---

[5]Data augmentation is considered as costly because it leads to (i) an increase of the number of training data, and/or (ii) an increase of the required number of training epochs.

Regarding future work, it could be interesting to tailor B-CNNs for segmentation tasks, given their relevance in fields such as biomedical and satellite imaging. Such domains benefit greatly from rotation and reflection equivariant models, making B-CNNs a promising candidate for these tasks. Next to that, it could also be interesting to further study the impact of the model sizes on the final performances for the different techniques. Indeed, as the different techniques bring the equivariance through different ideas, some of them are maybe more suitable than others for very small network sizes. Finally, a major actual concern in deep learning is the robustness regarding adversarial attacks or, more generally, small perturbations in the image. It could be interesting to evaluate if the use of Bessel coefficients and the $(S-)O(2)$ equivariant constraint make B-CNNs more robust to those specific perturbations or not.

## A   Appendix

In this Appendix we prove that the Bessel basis described in Section 4.1 can be used as an orthonormal basis by carefully choosing $k_{\nu,j}$.

**Theorem 1** *Let $D^2$ be a circular domain of radius $R$ in $\mathbb{R}^2$. Let $J_\nu(x)$ be the Bessel function of the first kind of order $\nu$, and let $k_{\nu,j}$ be defined such that $J'_\nu(k_{\nu,j}R) = 0$, $\forall \nu, j \in \mathbb{N}$. Then*

$$\left\{ N_{\nu,j} J_\nu(k_{\nu,j}\rho) e^{i\nu\theta} \right\}, \text{ where } N_{\nu,j} = 1 / \sqrt{2\pi \int_0^R \rho J_\nu^2(k_{\nu,j}\rho) \, d\rho}$$

*is an orthonormal basis well-defined to express any squared-integrable functions $f$ such that $f : D^2 \subset \mathbb{R}^2 \longrightarrow \mathbb{R}$.*

**Proof**   To prove this, we will use the fact that

$$\int_0^{2\pi} e^{i\theta(\nu'-\nu)} d\theta = \int_0^{2\pi} \cos(\theta(\nu'-\nu)) + i \int_0^{2\pi} \sin(\theta(\nu'-\nu))$$
$$= 2\pi\delta_{\nu,\nu'}, \tag{A.1}$$

since $\nu' - \nu$ is always an integer in our use of Bessel functions. We use also Lommel's integrals, which are in our particular case

$$\int_0^R \rho J_\nu(k_{\nu,j}\rho) J_\nu(k_{\nu,j'}\rho) \, d\rho =$$
$$\begin{cases} \frac{1}{k_{\nu,j'}^2 - k_{\nu,j}^2} \left[ \rho\left(k_{\nu,j} J'_\nu(k_{\nu,j}\rho) J_\nu(k_{\nu,j'}\rho) - k_{\nu,j'} J'_\nu(k_{\nu,j'}\rho) J_\nu(k_{\nu,j}\rho)\right) \right]_0^R & \text{if } k_{\nu,j} \neq k_{\nu,j'}, \\ \left[ \frac{\rho^2}{2} \left[ J'^2_\nu(k_{\nu,j}\rho) + \left(1 - \frac{\nu^2}{k_{\nu,j}^2\rho^2}\right) J_\nu(k_{\nu,j}\rho) \right] \right]_0^R & \text{otherwise.} \end{cases} \tag{A.2}$$

By taking into account that $J'_\nu(k_{\nu,j}R) = 0$, Lommel's integrals lead to

$$\int_0^R \rho J_\nu(k_{\nu,j}\rho) J_\nu(k_{\nu,j'}\rho) \, d\rho = \begin{cases} 0 \text{ if } k_{\nu,j} \neq k_{\nu,j'} \\ \left( \frac{R^2}{2} - \frac{\nu^2}{2k_{\nu,j}^2} \right) J_\nu(k_{\nu,j}R) & \text{otherwise.} \end{cases}$$
$$= \left( \frac{R^2}{2} - \frac{\nu^2}{2k_{\nu,j}^2} \right) J_\nu(k_{\nu,j}R) \delta_{j,j'}. \tag{A.3}$$

Now, by using Equation (A.1)

$$\int_0^{2\pi} \int_0^R \rho \left[ N_{\nu,j} J_\nu(k_{\nu,j}\rho) e^{i\nu\theta} \right]^* \left[ N_{\nu',j'} J_{\nu'}(k_{\nu',j'}\rho) e^{i\nu'\theta} \right] d\theta d\rho$$
$$= 2\pi\delta_{\nu,\nu'} \int_0^R \rho N_{\nu,j} J_\nu(k_{\nu,j}\rho) N_{\nu,j'} J_\nu(k_{\nu,j'}\rho) \, d\rho,$$

which, by using Equation (A.3), leads to

$$2\pi\delta_{\nu,\nu'} \int_0^R \rho N_{\nu,j} J_\nu(k_{\nu,j}\rho) N_{\nu,j'} J_\nu(k_{\nu,j'}\rho) \, d\rho$$
$$= 2\pi N_{\nu,j}^2 \left( \frac{R^2}{2} - \frac{\nu^2}{2k_{\nu,j}^2} \right) J_\nu(k_{\nu,j}R) \delta_{\nu,\nu'} \delta_{j,j'}.$$

To conclude this proof, one can show by using Equation (A.3) again that

$$N_{\nu,j}^2 = \frac{1}{2\pi \int_0^R \rho J_\nu^2(k_{\nu,j}\rho) \, d\rho}$$
$$= \frac{1}{2\pi \left( \frac{R^2}{2} - \frac{\nu^2}{2k_{\nu,j}^2} \right) J_\nu(k_{\nu,j}R)},$$

and then finally,

$$\int_0^{2\pi} \int_0^R \rho \left[ N_{\nu,j} J_\nu \left( k_{\nu,j} \rho \right) e^{i\nu\theta} \right]^* \left[ N_{\nu',j'} J_{\nu'} \left( k_{\nu',j'} \rho \right) e^{i\nu'\theta} \right] d\theta d\rho = \delta_{\nu,\nu'} \delta_{j,j'},$$

which is the definition of an orthonormal basis[6].

## B  Appendix

In this Appendix we prove the properties that link $\varphi_{\nu,j}$ and $\varphi_{-\nu,j}$.

**Theorem 2** *Let $\varphi_{\nu,j}, \forall \nu, j \in \mathbb{N}$ be the Bessel coefficients of a particular function $\Psi\left(\rho, \theta\right) : D^2 \subset \mathbb{R}^2 \longrightarrow \mathbb{R}$ defined on a circular domain of radius $R$, that is,*

$$\varphi_{\nu,j} = \int_0^{2\pi} \int_0^R \rho \left[ N_{\nu,j} J_\nu \left( k_{\nu,j} \rho \right) e^{i\nu\theta} \right]^* \Psi\left(\rho, \theta\right) d\theta d\rho.$$

*Then, these coefficients are not all independent. They are linked by the relations*

$$\varphi_{-\nu,j} = \left(-1\right)^\nu \varphi_{\nu,j}^* \iff \begin{cases} \Re\left(\varphi_{-\nu,j}\right) = \left(-1\right)^\nu \Re\left(\varphi_{\nu,j}\right) \\ \Im\left(\varphi_{-\nu,j}\right) = \left(-1\right)^{\nu+1} \Im\left(\varphi_{\nu,j}\right). \end{cases}$$

**Proof**  To prove this, we will use different properties of the Bessel functions. Firstly,

$$J_{-\nu}\left(x\right) = \left(-1\right)^\nu J_\nu\left(x\right),$$

and secondly,

$$J_\nu'\left(x\right) = \frac{1}{2} \left( J_{\nu-1}\left(x\right) - J_{\nu+1}\left(x\right) \right).$$

Then, by using these two relations, one can show that

$$\begin{aligned} J_{-\nu}'\left(x\right) &= \frac{1}{2} \left( J_{-\nu-1}\left(x\right) - J_{-\nu+1}\left(x\right) \right) \\ &= \frac{\left(-1\right)^{\nu+1}}{2} \left( J_{\nu+1}\left(x\right) - J_{\nu-1}\left(x\right) \right) \\ &= \left(-1\right)^\nu J_\nu'\left(x\right) \end{aligned} \tag{B.1}$$

However, if $k_{-\nu,j}$ is such that $J_{-\nu}'\left(k_{-\nu,j}R\right) = 0$, it also leads thanks to Equation (B.1) to $J_\nu'\left(k_{-\nu,j}R\right) = 0$. And then, the only possibility is that $k_{-\nu,j} = k_{\nu,j}$ (because we still have $J_\nu'\left(k_{\nu,j}R\right) = 0$).

Now, regarding the normalization factor,

$$\begin{aligned} N_{-\nu,j}^{-1} &= \sqrt{2\pi \int_0^R \rho J_{-\nu}^2 \left( k_{-\nu,j} \rho \right) d\rho} \\ &= \sqrt{2\pi \int_0^R \rho J_\nu^2 \left( k_{\nu,j} \rho \right) \left(-1\right)^{2\nu} d\rho} \\ &= N_{\nu,j}^{-1}. \end{aligned}$$

---

[6]Note that the proof for $k_{\nu,j}$ defined by $J_\nu\left(k_{\nu,j}R\right) = 0$ is now straightforward since it only sweeps the non-zero term in Equation (A.2). The following remains the same.

One can now put all this together to show that

$$\varphi_{-\nu,j} = \int_0^{2\pi} \int_0^R \rho \left[ N_{-\nu,j} J_{-\nu} \left( k_{-\nu,j} \rho \right) e^{-i\nu\theta} \right]^* \Psi \left( \rho, \theta \right) d\theta d\rho$$

$$= (-1)^\nu \int_0^{2\pi} \int_0^R \rho \left[ N_{\nu,j} J_\nu \left( k_{\nu,j} \rho \right) e^{-i\nu\theta} \right]^* \Psi \left( \rho, \theta \right) d\theta d\rho$$

$$= (-1)^\nu \varphi_{\nu,j}^*,$$

which leads to the end of the proof. ∎

## C  Appendix

In this Appendix, we prove that the rotation invariant operation described in Section 5.1 is pseudo-injective. It means that results will be different if images are different. Pseudo makes reference to the exception when an image is compared to a rotated version of itself.

**Theorem 3** *Let $\{\varphi_{\nu,j}\}$ be the Bessel coefficients of a particular function $\Psi \left( \rho, \theta \right) : D^2 \subset \mathbb{R}^2 \longrightarrow \mathbb{R}$ defined on a circular domain of radius $R$. Let $\{\varphi'_{\nu,j}\}$ be the Bessel coefficients of another particular function $\Psi' \left( \rho, \theta \right)$ defined on the same domain $D^2$. Finally, let $\{\kappa_{\nu,j}\}$ be some arbitrary complex numbers. Then,*

$$\sum_\nu \Big| \sum_j \kappa_{\nu,j}^* \varphi_{\nu,j} \Big|^2 = \sum_\nu \Big| \sum_j \kappa_{\nu,j}^* \varphi'_{\nu,j} \Big|^2 \Rightarrow \exists \alpha : \Psi \left( \rho, \theta \right) = \Psi' \left( \rho, \theta - \alpha \right), \forall \rho, \forall \theta.$$

**Proof**  To make developments easier, one can use the bra-ket notation commonly used in quantum mechanics to denote quantum states. In this notation, $|v\rangle$ is called a *ket* and denotes a vector in an abstract complex vector space, and $\langle v|$ is called a *bra* and corresponds to the same vector but in the dual vector space. It follows that $|v\rangle^\dagger = \langle v|$, and the inner-product between two vectors is conveniently expressed by $\langle v|f\rangle$, and the outer-product by $|f\rangle\langle v|$.

By using the fact that $|z|^2 = zz^*$ and the bra-ket notation,

$$\sum_\nu \Big| \sum_j \kappa_{\nu,j}^* \varphi_{\nu,j} \Big|^2 = \sum_\nu \Big| \sum_j \kappa_{\nu,j}^* \varphi'_{\nu,j} \Big|^2$$

leads to

$$\sum_\nu \langle \kappa_\nu | \varphi_\nu \rangle \langle \kappa_\nu | \varphi_\nu \rangle^* = \sum_\nu \langle \kappa_\nu | \varphi'_\nu \rangle \langle \kappa_\nu | \varphi'_\nu \rangle^*,$$

where $\kappa_\nu$ (resp., $\varphi_\nu$) is a vector that contains all the different values $\kappa_{\nu,j}$ (resp., $\varphi_{\nu,j}$) for this particular $\nu$. This Equation can further be written

$$\sum_\nu \langle \kappa_\nu | \varphi_\nu \rangle \langle \varphi_\nu | \kappa_\nu \rangle = \sum_\nu \langle \kappa_\nu | \varphi'_\nu \rangle \langle \varphi'_\nu | \kappa_\nu \rangle. \tag{C.1}$$

However, since the $\kappa_{\nu,j}$'s are totally arbitrary, the only possibility to satisfy Equation (C.1) is that

$$\sum_\nu | \varphi_\nu \rangle \langle \varphi_\nu | = \sum_\nu | \varphi'_\nu \rangle \langle \varphi'_\nu |.$$

In quantum mechanics, $|\varphi_\nu\rangle\langle\varphi_\nu|$ is called the density matrix of $\varphi_\nu$, and it is known that the only way to achieve identical density matrices for different states $\varphi_\nu$ and $\varphi'_\nu$ is that they should only differ by a phase factor[7]. ∎

---

[7]Indeed, if $\varphi_\nu = \varphi'_\nu e^{i\nu\alpha}$, then $|\varphi'_\nu\rangle\langle\varphi'_\nu| = |e^{i\nu\alpha}\varphi_\nu\rangle\langle e^{i\nu\alpha}\varphi_\nu| = e^{i\nu\alpha}e^{-i\nu\alpha}|\varphi_\nu\rangle\langle\varphi_\nu| = |\varphi_\nu\rangle\langle\varphi_\nu|$

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
