# OpenReview forum: "SO(2) and O(2) Equivariance in Image Recognition with Bessel-Convolutional Neural Networks"
_TMLR — Rejected by TMLR_

### Review · Reviewer_hZTS · 2024-10-21

**Summary Of Contributions:**

The paper studies Bessel-convolutional neural networks (B-CNNs), which replace ordinary convolutional layers by specific rotation equivariant convolutions. Convolution filters are parameterized radially by Bessel functions and angularly by circular harmonics. A comprehensive background on B-CNNs is given with more details than prior work and the framework of B-CNNs is extended to the settings of reflection and scale equivariance. Experiments are carried out in the small dataset regime, showing the potential benefit of B-CNNs there.

**Audience:**

Yes

**Broader Impact Concerns:**

-

**Claims And Evidence:**

No

**Requested Changes:**

### Critical for recommending acceptance
1. It is claimed on page 13 and in Appendix C, that the proposed convolution is "pseudo-injective", i.e. gives a different output for any inputs that are not related by a rotation. I am not convinced that this is correct. In the proof on page 42, it is shown that the frequency components $\varphi\_\nu$ and $\varphi'\_\nu$ can differ by phase. The authors conclude that the functions $\Psi$ and $\Psi'$ must be related by a phase shift, but as far as I understand, the phase shift can be different for each frequency component $\nu$, meaning that there is not necessarily a global phase shift aligning $\Psi$ and $\Psi'$. Intuitively, taking the energy (sum over squared modulus) of a function $f$ will ignore the relative phases of the frequency components of $f$, so the proposed operation should lose more information than just one "global" rotation.
2. The graphs for the experimental results are quite difficult to read. For example, in Fig 12, dark blue should correspond to a standard CNN, the most basic baseline, but I cannot make out a dark blue line in any of the top row graphs. I have similar issues with many other graphs. Including fewer methods here and moving the current graphs to the appendix would be helpful.

### Not critical
1. The datasets considered are small. Not even complete MNIST is considered, but only up to 20% of MNIST data. It is not entirely clear why full MNIST is omitted.
2. The theoretical comparison with general E(2)- steerable CNNs could be more in-depth. The main differences, as I understand them, are the choice of radial profile for the filters and the fact that in E(2)-steerable CNNs, features of irrep-type > 0 are kept between layers, whereas in B-CNN, the irrep-types are aggregated to irrep-type 0 by summing the squared moduli after each convolution. Is this understanding correct?
3. If I'm not mistaken, the derivation leading up to eq (5.6) could be shortened by referring to Parseval's theorem. I.e. the integral of the squared modulus in the spatial domain (5.2) equals the sum over the squared frequency spectrum in (5.6).

**Strengths And Weaknesses:**

The paper is well-written and could become a standard reference for Bessel-convolutional neural networks (B-CNNs). There are some weaknesses that can be easily fixed, see Requested Changes. The experiments are reasonable for the small dataset regime, but perhaps not convincing for using B-CNNs in practice on larger datasets.

---

### Review · Reviewer_RRXJ · 2024-11-09

**Summary Of Contributions:**

This manuscript is an extension of Delchevalerie et al. 2021, published at NeurIPS 2021, where the use of Bessel functions as basis functions for images is studied in order to obtain activations with a predictable behavior wrt rotations of the input, and thus allow for rotation equivariance. The extension consists mainly in going deeper in some of the explanations. In addition, the addition of reflection invariance is also explored, as suggested by Delchevalerie et al., as well as scale invariance via an image pyramid.

**Audience:**

No

**Broader Impact Concerns:**

I do not think this submission presents any specific ethical implications.

**Claims And Evidence:**

Yes

**Requested Changes:**

In case that I may have missed some important element in the paper that would make the authors think that this manuscript is not an extension of Delchevalerie et al. but a fully original submission, I think such element should be clearly put forward in a "Contributions" section.

**Strengths And Weaknesses:**

The manuscript follows quite closely the structure, wording and notation of Delchevalerie et al. I find many of the additions quite pedagogical, and I do think that the more detailed explanations can help better understand the method. I could not find any wrong nor misleading statements beyond what is exposed in the following.

On the other hand, the methodological additions wrt Delchevalerie et al. are very limited:
1) The addition on image reflections represents, if I understand it well, an invariant behaviour rather than equivariant, since it consists of zeroing the terms that contain the reflection information. This loss of information could explain the fact that adding this invariance to the model results in degraded performances.
2) Similarly, the multi-scale aspect is implemented via max-pooling over several scale representations, thus also encoding scale invariance rather than equivariance.
The advantage of these additions seems to be limited, according to the presented experiments, and serve rather to complete the work of Delchevalerie et al. rather than being of interest to readers per se.

Overall, I don’t think this manuscript can be considered an original contribution, but rather as an extension of Delchevalerie et al. Since the extensions of conference articles are not considered within the scope of TMLR (as per the Submission Guidelines and Editorial Policies), I would suggest that the authors may want to try with a different venue.

---

### Review · Reviewer_wem1 · 2024-11-23

**Summary Of Contributions:**

This paper presents Bessel Convolutional Neural Networks (B-CNNs), where the convolutional kernels are learned on the basis of Bessel functions. The authors present a formulation in which a *weak* form --see weaknesses-- of rotation, mirroring and scale equivariance. This paper is n extension of the paper:

> Valentin Delchevalerie, Adrien Bibal, Benoît Frénay, and Alexandre Mayer. Achieving rotational invariance with bessel-convolutional neural networks. In Advances in Neural Information Processing Systems (NeurIPS), pp. 28772–28783, 2021.

With some additional contributions:

* The optimal selection of $v_\mathrm{max}$ and $j_\mathrm{max}$ --which decide the size of the basis to use-- based on the Nyquist criterion.
* Extension to mirroring.
* Extension to a *weak* form of equivariance to scaling.

**Audience:**

Yes

**Claims And Evidence:**

No

**Requested Changes:**

Please see the previous comments.

**Strengths And Weaknesses:**

## Suitability for TMLR.

Before delving onto the review as such, I am afraid I am not sure this is the right venue for this paper. As stated in https://jmlr.org/tmlr/editorial-policies.html, this journal is intended for original work only:

> Unlike many other journals, TMLR only accepts original contributions that don’t reuse the authors’ own prior work. In particular, **we do not accept submissions that are expanded versions of conference papers**. There should not be any reuse of written text, figures or results between the submitted paper and any paper which has been published, accepted for publication, or submitted in parallel at another archival, peer-reviewed venue.

Based on this, I am not sure this is the right venue for this paper. However, given that I did go through the paper, I will give my review nonetheless. I leave it to the Action Editor to decide whether this submission should be desk rejected.

---

## Main Weaknesses

- The principal weakness of this paper is regarding to its claims. I would like to call to the attention of the authors that the equivariance obtained by this work is **a weaker form of equivariance than that used in other works**. This is true for all the groups considered. In particular, note that method is equivariant at a global scale but **invariant** at a local scale. This can be clearly observed in Fig. 11. As seen here, the method uses max pooling along scales to give the final answer. Consequently, subsequent layers won't be able to know which scale was actually observed in the input. This is different to the formulation of most equivariant methods, such as G-CNNs, E2-CNNs, etc, whose operations **keep** this information on the output as well, encoded on the additional group dimension.

  Now, this difference is by not clear nor specified in the paper. It is assumed that the this method achieves the same type of equivariance as the other works. This difference should be made extremely clear in the paper.

- The paper starts by defining how the input and the conv kernels can be defined on the Bessel basis and how convolutions can be described on that space as well. However, at the end, as depicted in Page 17, the method becomes a normal convolution where the only differences are that (i) the conv kernels are defined on the Bessel basis, and (ii) an square operation is used after convolution --which is responsible for the local invariance by removing phase information--. This presentation feels tedious and unnecessary. It would have been much easier to describe the method as describing the kernels in a continuous basis as opposed to a discrete basis, as done, e.g., in Bekkers et al 2019.

- The authors claim that their method, in contrast to existing SO2-equivariant methods, allows their method to be compatible with existing deep learning methods (Page 13). This is not true due to two reasons. First, as mentioned later in the paper, B-CNNs do not work well neither with batchnorm, nor ReLU, which directly contradicts the claim. In addition, there exist methods such as Finzi et al 2021, which are equivariant to continuous groups, yet are able to use all existing methods (batchnorm, relu, etc). In fact, these works have been later on extended to Sim(2) as well as O(2), while considering a full form of equivariance --as opposed to the method here presented (Knigge et al 2021, Romero & Lohit, 2022). In terms of comparisons to other methods, I feel that these methods --which are both continuous and use regular representations-- would offer better comparisons to B-CNNs. For example, in Page 18, the authors discuss how E2-CNNs use n replicas for $C_n$ groups, yet, it is possible with these methods to obtain continuous equivariance, while using, e.g., 4 group elements.

- Finally, the whole of the experimental section is based on datasets with are very simple. Note how all models are most of the time above 95% accuracy in all settings. This makes it difficult to gauge the gains --or constraints-- of using B-CNNs. I strongly recommend to the authors to evaluate on more difficult datasets and with more complicated architectures (See the architectures used in Table 2).

### Other weaknesses

- This paper is extremely long (44 pages), especially considering that the contributions are relatively small compared to Delchevalerie et al. I would strongly advice the authors to restructure the paper and reduce the length of the main text. Many parts could safely be sent to the appendix.

- Note that similar analyses on the Nyquist criteria have been used broadly in several works to restrict the size of the bases used. This has been both in the context of equivariance, as well as more broadly in deep learning and beyond. The authors should make this clear in the paper and add the corresponding references. Given that many of these concepts are known, e.g., aliasing, a big chunk of this part could safely be sent to the appendix.

- The introduction has many claims, but does not include references. Each claim,e.g., "... most of these methods do not provide any mathematical guarantee regarding equivariance", should be supported by references or explanations.

- On the same vein, the authors claim that "a few works propose solutions to efficiently bring more general equivariances in CNNs while providing mathematical guarantees." -> This is not true. I can easily name over 50 papers that do this.

- I observe some confusions in the related work section. Note that Finzi et al achieve equivariance to the continuous group while using regular representations. In general, regular representations consistently lead to better results than other representations such as irreducible representations.

- This work is also related to steerable functions. A discussion on the differences between bases used, e.g., Weiler et al 2018; 2019, would be valuable.

- The discussion between discrete vs continuous groups in Page 37 is outdated. Note how there exist many methods that are continuous and use regular representations.

- Given the scope, goals and experimental section of this work, the discussion on Dehmamy et al feels disconnected from the rest of the paper. This should probably be on the Future Work section.

### Conclusion

Based on my previous comments, I, unfortunately, cannot recommend acceptance at this point. The paper needs improvements and clarifications in several fronts. In addition, I am not sure whether TMLR is the right venue for this paper, as explain previously.

---

### Decision · Action_Editor_d6Ge · 2025-01-06

**Recommendation:** Reject

**Comment:**

As noted above, there is a strong consensus that this work must be viewed as an extension of Delchevalerie et al. (NeurIPS 2021) rather than a new work.  The TMLR guidelines are clear that extensions of full-length conference papers are not appropriate, so this issue alone appears to be sufficient grounds for rejection.  In addition, the reviewers highlighted a number of concerns and questions but did not receive any response.

**Audience:**

The paper topic itself is certainly suitable to TMLR, as it centers around deep learning.  However, some important concerns are raised, as outlined below.

**Claims And Evidence:**

This paper studies equivariances in imaging problems via Bessel Convolutional Neural Networks (B-CNNs), in which convolutional kernels are learned via Bessel functions.  The proposed formulation can naturally encode certain forms of rotation/mirroring/scale invariances.  The main claims include covering more invariances compared to existing works, more detailed mathematical analyses, and extensive experimental results.  However, there was a strong consensus that this work must be viewed as an extension of Delchevalerie et al. (NeurIPS 2021).